# SynFlowNet: Design of Diverse and Novel Molecules with Synthesis Constraints

**Miruna Cretu[1], Charles Harris[1], Ilia Igashov[2], Arne Schneuing[2], Marwin Segler[3],
Bruno Correia[2], Julien Roy[4], Emmanuel Bengio[4], Pietro Liò[1]**
[1]University of Cambridge, [2]EPFL, [3]Microsoft Research, [4]Valence Labs
`{mtc49, cch57, pl219}@cam.ac.uk`,
`emmanuel.bengio@recursionpharma.com`

## Abstract

Generative models see increasing use in computer-aided drug design. However, while performing well at capturing distributions of molecular motifs, they often produce synthetically inaccessible molecules. To address this, we introduce SynFlowNet, a GFlowNet model whose action space uses chemical reactions and purchasable reactants to sequentially build new molecules. By incorporating forward synthesis as an explicit constraint of the generative mechanism, we aim at bridging the gap between in silico molecular generation and real world synthesis capabilities. We evaluate our approach using synthetic accessibility scores and an independent retrosynthesis tool to assess the synthesizability of our compounds, and motivate the choice of GFlowNets through considerable improvement in sample diversity compared to baselines. Additionally, we identify challenges with reaction encodings that can complicate traversal of the MDP in the backward direction. To address this, we introduce various strategies for learning the GFlowNet backward policy and thus demonstrate how additional constraints can be integrated into the GFlowNet MDP framework. This approach enables our model to successfully identify synthesis pathways for previously unseen molecules. Source code is available at `https://github.com/mirunacrt/synflownet`.

## 1 Introduction

Designing molecules with targeted biochemical properties is a critical challenge in drug discovery, where computational models could play a significant role to increase efficiency and effectiveness. Recently, generative models have lead to a renaissance in *de novo* molecular design (Stanley & Segler, 2023; Du et al., 2024). However, most current *de novo* design models do not explicitly account for synthetic accessibility. Many approaches operate on SMILES strings (Segler et al., 2017; Gupta et al., 2018) or assemble molecules by composing atoms or molecular fragments into a graph (Lewis & Dean, 1989; Jensen, 2019; Jin et al., 2018), leaving no guarantee that the sampled molecules can be synthesized (Stanley & Segler, 2023; Gao & Coley, 2020).

To address this limitation, synthetic complexity scores (Ertl & Schuffenhauer, 2009; Coley et al., 2018; Liu et al., 2022) have been proposed as a method to assess molecules and complement generative models with knowledge on synthetic accessibility, however these heuristics and learned metrics are often oversimplified. Alternatively, computer-aided synthesis planning can can be performed as a subsequent step to molecule generation (Segler et al., 2018; Schwaller et al., 2020). While synthesis planning can provide principled routes, it can take seconds to minutes to propose pathways, which hampers its integration as a reward function in the generation process (Liu et al., 2022). Finally, synthetically accessible chemical spaces can be assembled from combinations of reactions involving readily available reactants (Klarich et al., 2024), but virtual screening on these vast datasets is already impractical today (screening only a fraction of such libraries amounts to thousands of years of compute on a single CPU; Sadybekov et al., 2022) and these spaces are still growing exponentially.

Given the challenges of synthesizability in atom- and fragment-based generative models, we propose formulating molecule generation in an action space of chemical reactions which naturally enhances synthesizability. However, simply employing a synthesizable space is not sufficient. To be applicable

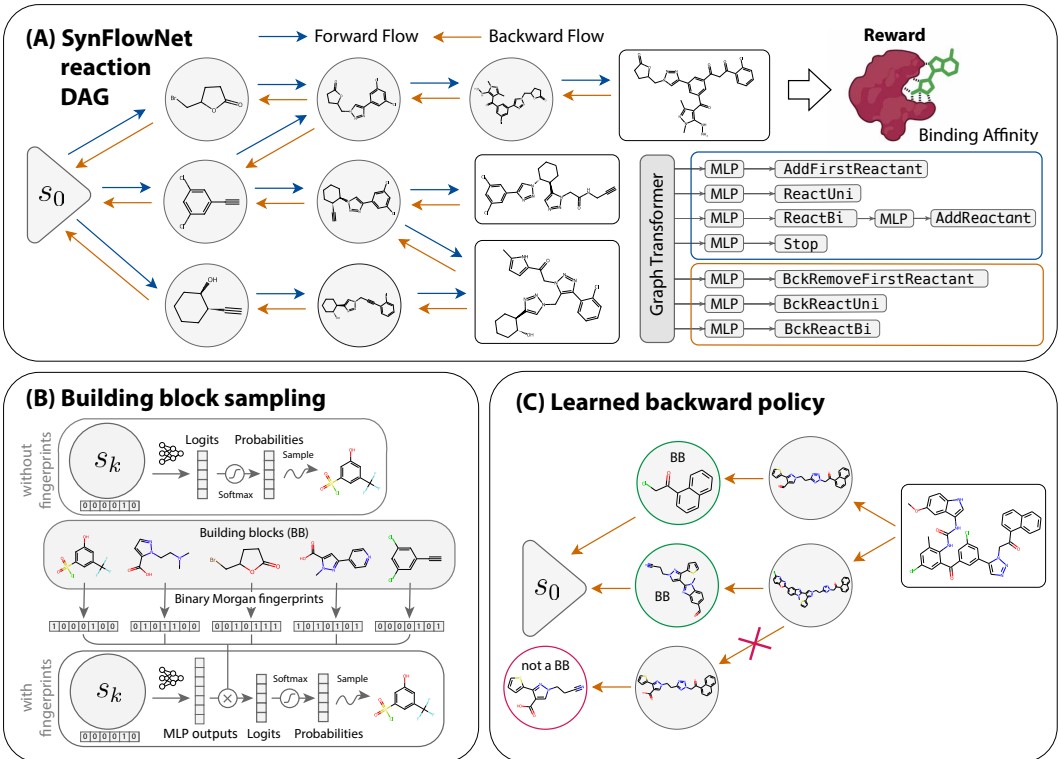

Figure 1: **SynFlowNet allows for synthesis-aware molecule generation.** (A) The state space is induced by combining purchasable building blocks (BBs) and chemical reactions. Every final molecule (rectangle box) is associated with a reward. Training trajectories are constructed by sampling the model forward. Our policy $P_F(a|s)$ is parameterized as a graph transformer which at each timestep processes the current molecular state $s_t$ and outputs a shared embedding which is passed to separate MLP heads to predict the action logits for different action types (5 forward and 3 backward action types). An action $a_t$ is then sampled from this hierarchical distribution to transition to the next state via a reaction. (B) To allow handling large sets of BBs (up to 200k), we represent them using Morgan fingerprints and compute the probability of sampling a particular BB from the normalised dot product between this representation and the MLP output for the current state. The state representation is concatenated with the one-hot encoding of the selected reaction. (C) Finally, when traversing the MDP backwards, to reduce the probability of exiting the MDP defined by our set of reactions and BBs, we train the backward policy to avoid paths that do not terminate in $s_0$.

to drug discovery programs, such a model needs to maximize desired properties and retrieve molecules from distinct modes of the available synthetic space. Generative Flow Networks (GFlowNets; Bengio et al., 2021) have emerged as a framework capable of generating diverse samples and requiring fewer evaluations of the reward function compared to alternatives like Markov Chain Monte Carlo or Proximal Policy Optimization (Schulman et al., 2017).

In this work, we introduce SynFlowNet, a GFlowNet specifically trained to generate molecules from documented chemical reactions and purchasable starting materials, thereby constraining exploration to a synthetically accessible chemical space and sampling not only target compounds but also the synthetic routes leading to them. Our main contributions are:

- We train a GFlowNet using an action space defined by documented chemical reactions and purchasable starting materials to generate synthesizable molecules.

- We show the advantage of using a reaction-based environment over a fragment-based one in terms of synthesisability, and the benefits of GFlowNets over Reinforcement Learning (RL) in terms of sample diversity on several targets. Additionally, when comparing to other generative methods with synthetically-accessible outputs, we show that SynFlowNet generates more novel candidates w.r.t known drug-like molecules.

- We evaluate different action representation alternatives to allow efficient scaling of the action space to up to 200k chemical building blocks.

- We identify an inherent problem of employing a reaction-based Markov Decision Process (MDP) with GFlowNets, stemming from the lack of guarantee that backward-constructed trajectories can return to the initial state $s_0$. To resolve this issue, we propose one of the first attempts at training the backward policy in a GFlowNet with a separate objective from the forward policy, which we show to correct backward-generated trajectories in the MDP while retaining the ability to discover diverse and high-reward modes.

- We show that our proposed framework can be integrated with target-specific experimental data to inform selection of building blocks, which improves the efficiency of our model.

## 2 BACKGROUND AND RELATED WORK

### 2.1 GFLOWNETS

GFlowNets (Bengio et al., 2021) are a class of probabilistic models that learn a stochastic policy to generate objects $x$ through a sequence of actions, with probability proportional to a reward $R(x)$. The sequential construction of objects $x$ can be described as a trajectory $\tau \in \mathcal{T}$ in a directed acyclic graph (DAG) $G = (S, \mathcal{E})$, starting from an initial state $s_0$ and using actions $a$ to transition from a state to the next: $s \rightarrow s'$. A GFlowNet uses a forward policy $P_F(-|s)$, which is a distribution over the children of state $s$, to sample a sequence of actions based on the current states. Similarly, a backward policy $P_B(-|s)$ is the distribution over the parents of state $s$, and can be used to calculate probabilities of backward actions, leading from terminal to initial states. The training objective which we adopt in this paper is trajectory balance (Malkin et al., 2022):

$$\mathcal{L}_{\text{TB}}(\tau) = \left( \log \frac{Z_\theta \prod_{t=1}^{T} P_F(s_{t+1}|s_t; \theta)}{R(x) \prod_{t=1}^{T} P_B(s_{t-1}|s_t; \theta)} \right)^2 \tag{1}$$

used to learn the forward and backward policies $P_F(-|s; \theta)$ and $P_B(-|s; \theta)$ parameterized by $\theta$ and to estimate the partition function $Z_\theta \approx F(s_0) = \sum_{\tau \in \mathcal{T}} F(\tau)$.

Bengio et al. (2021) have used GFlowNets to generate molecules with high binding affinity to a protein target by linking fragments to form a junction tree (Jin et al., 2019). For generative chemistry, the framework was extended to multi-objective optimisation (Jain et al., 2023; Roy et al., 2023), where the model was trained to simultaneously optimise for binding affinity to the protein target, Synthetic Accessibility (SA), drug likeness (QED) and molecular weight.

### 2.2 GFLOWNETS WITH A PARAMETERIZED BACKWARD POLICY

GFlowNets train a forward policy $P_F$ to match the backward policy $P_B$ according to Eq. 1. The choice of $P_B$ therefore impacts the training of GFlowNets and sample quality. Despite this relationship, the choice of backward policy in GFlowNets has attracted limited attention but for a few works discussed here. Malkin et al. (2022) proposed parameterizing the backward policy and training $P_B$ and $P_F$ simultaneously using the trajectory balance objective (Eq. 1). They also proposed fixing the backward policy to a uniform distribution when modeling the distribution over parents states proves difficult. Mohammadpour et al. (2024) compare a maximum-entropy GFlowNet to GFlowNets with a uniform backward policy. Closer to our work, Jang et al. (2024) propose Pessimistic GFlowNets, which use *maximum likelihood* over observed trajectories to train $P_B$. This ensures that the flow induced by the backward policy is concentrated around observed (training) states, making the model pessimistic about unobserved intermediate states having flow. In this work, we address the idea of ensuring that backward-generated trajectories belong to the GFlowNet MDP. In doing so, we propose a data-driven solution to a MDP design challenge.

### 2.3 GENERATIVE MODELS FOR MOLECULE DESIGN

A large number of works have been proposed for generative molecular design (Du et al., 2024). Early methods employed techniques such as variational autoencoders (VAEs) (Gómez-Bombarelli et al., 2018), deep reinforcement learning (Segler et al., 2017; Olivecrona et al., 2017b), and generative

adversarial networks (GANs) (Guimaraes et al., 2018; Cao & Kipf, 2022). (Zang & Wang, 2020) Despite these advances, challenges remain, particularly in ensuring that generated molecules adhere to physical and chemical constraints (Stanley & Segler, 2023; Harris et al., 2023).

## 2.4 Synthesis-aware molecule generation

The idea of tackling molecule generation and synthesis simultaneously has been investigated early by Vinkers et al. (2003), who introduced SYNOPSIS, which generates molecules from a starting dataset of available compounds, relying on applying chemical modifications to functional groups and assessing the value of the product with a fitness function. The works of Bradshaw et al. (2019) and Korovina et al. (2020) provided early neural models for one-step synthetic pathways. Bradshaw et al. (2020) generalized this idea as a generative model for synthesis DAGs, which can be optimized in a VAE or RL setup. Gottipati et al. (2020b) used reinforcement learning to generate compounds from reactions and commercially available reactants. Gao et al. (2021) formulate an MDP to model the generation of synthesis trees, which can be optimized with respect to the desired properties of a product molecule. Luo et al. (2024) proposed a model that can project unsynthesizable molecules from existing generative models to synthesizable chemical space by utilizing postfix notations to represent synthesis pathways. Guo & Schwaller (2024) showed that retrosynthesis models can be treated as an oracle in goal-directed molecule generation. Concurrently to our work, Koziarski et al. (2024) propose a similar GFlowNet-based framework for synthesizable molecular generation but using a different set of reaction templates and making use of a different backward policy.

While our work resembles an RL setting for synthesis-aware molecular generation (Gottipati et al., 2020b; Horwood & Noutahi, 2020), the key difference lies in the sampling distribution of the learned model. Contrary to RL, the GFlowNet objective is not to generate the single highest-return sequence of actions, but rather to maximise both performance and diversity by sampling terminal states proportionally to their reward. This is especially useful in the context of molecule generation, where we want to explore different modes of the distribution of interest.

## 3 Methods

In this work, we present a framework to train GFlowNets on a Markov Decision Process (MDP) made of molecules obtained from sequences of chemical reactions. Below, we describe how this compositional space of synthesizable molecules is assembled (3.1), we present a method making use of backward policies to palliate imperfect information contained in reaction templates (3.2) and present the model used to navigate that state space and learn the target distribution (3.3).

### 3.1 MDP of chemically accessible space

**Problem definition**   We model synthetic pathways as trajectories in a GFlowNet, starting from purchasable compounds and ending with molecules that are optimized for some desired properties, via a set of permissible reaction templates. At each timestep $t$, the state $s_t$ represents the current molecule and stepping forward in the environment consists in building up the molecule by applying new pairs of reactions and reactants until either a termination action is chosen or the path reaches a maximum length. We encode reaction templates using RDKit reaction SMARTS (see Figure A.1).

**Forward Action Space**   We define five types of forward actions: `Stop`, `AddFirstReactant`, `ReactUni`, `ReactBi`, and `AddReactant`. `ReactUni` and `ReactBi` represent uni-molecular and bi-molecular reactions. The `AddReactant` action, which is available only after a `ReactBi` action, represents the choice of reactant for the bi-molecular reaction. In more detail, each trajectory starts from an empty molecular graph which is followed by a building block sampled from `AddFirstReactant`. We then continue based on the sampled action type as follows: (a) if the action type is `Stop`, we reach a terminal state and end the trajectory; (b) if a `ReactUni` action is sampled, we apply the uni-molecular reaction template to the molecule in state $s$ to yield the product in state $s'$; (c) if the action type is `ReactBi`, the sampled reaction is used as input to an additional MLP, together with the state embedding, to sample a subsequent action of type `AddReactant`.

**Backward Action Space**  The GFlowNet framework requires us to model travelling *backward* along trajectories in the state-space. To unfold a reverse trajectory we proceed similarly: (a) if the action-type is a `BckReactUni`, the action yields the reactant molecule directly; (b) if the action type is `BckReactBi`, we obtain two reactants, and the molecule that is **not** a building block becomes the *next* state (or *previous* state in the DAG). If the two resulting reactants are both building blocks (which happens at the beginning of the forward trajectory), the molecule that populates the next state is picked with $p = 1/2$ from the two building blocks. The last action in a backward trajectory is `BckRemoveFirstReactant`, leading to the empty molecular graph $s_0$ (initial state).

**Masking**  Prior to sampling actions in both forward and backward direction, we ensure that the reactions and building blocks to be sampled are compatible with the current state using masks, obtained by checking for substructure match between the reaction template and the reactant (for forward actions) or product (for backward actions) molecules. We also enforce through masking that at least one of the resulting reactants when running a backward reaction is a building block.

### 3.2 CHALLENGES WITH GENERATING BACKWARD PATHS

Given the synthesis pointed DAG $\mathcal{G} = (\mathcal{S}, \mathcal{A})$, one needs to define a forward probability function $P_F$ and a backward probability function $P_B$ both consistent with $\mathcal{G}$ (see Eq. 1). Contrary to previous fragment-based or atom-based molecule-generation GFlowNet environments, where any backward action can lead to $s_0$ (removing nodes and edges sequentially will lead to an empty graph; Bengio et al., 2021), defining $P_B$ in a reaction-based environment is non-trivial. This is because not every parent state (obtained by applying a reaction template backwards) will ensure that there exists a sequence of actions that leads all the way back to a building block, and therefore $s_0$. Note that the masking described in Section 3.1 is insufficient to account for this, as it does not ensure that the state obtained is *further* decomposable into building blocks. Consequently, to maintain a *pointed* DAG, no flow should be assigned to such a transition. A uniform backward policy, which is a standard choice in GFlowNet literature (Malkin et al., 2022), will fail at achieving this as it will assign positive flow to every backward action, including those leading to states that are not attainable from forward trajectories initialised in $s_0$ (see Figure 1-C). To address this issue, we explore a few training schemes for a parameterized $P_B$ that force backward-constructed trajectories to end in $s_0$.

**Training the backward policy**  We first explore a training scheme for $P_B$ which makes use of forward-generated trajectories. Similarly to Jang et al. (2024), we train $P_B$ using the maximum likelihood objective over trajectories generated from $P_F$:

$$\mathcal{L}_B(\theta_B) = \mathbb{E}_{\tau \sim P_F}[-\log P_B(\tau; \theta_B)]. \tag{2}$$

In that setting, we (1) generate trajectories using $P_F$, (2) update $P_F$ according to the trajectory balance objective in Equation 1 and (3) update $P_B$ using these same trajectories according to Equation 2 (see Algorithm 1 in Appendix A.4).

While the maximum-likelihood approach presented above is sufficient to limit $P_B$ in allocating flow to paths that do not connect back to $s_0$, it restricts the model's exploration by encouraging $P_F$ to collapse on a single path for each terminal molecule. To allow $P_B$ to ban erroneous paths while retaining a higher entropy, we also explore the use of policy gradient methods. Specifically, we explore maximising an expected *backwards* reward via REINFORCE (Williams, 1992), which is suitable for short trajectory environments like our reaction-based MDP:

$$J_B(\theta_B) = E_{\tau \sim P_F, P_B}[R_B(\tau)] - \alpha H(P_B). \tag{3}$$

Here, $H(P_B) = -\mathbb{E}_{\tau \sim P_B}[\log(P_B(\tau))]$ is the entropy term and the reward $R_B$ is set to 1 for a trajectory that ends in $s_0$ and -1 otherwise. In this setting we train the backward policy not only on the trajectories generated by the forward policy, but also on newly generated backward trajectories sampled directly from $P_B$ (see Algorithm 2, Appendix A.4).

Interestingly, training the backward policy to navigate back to $s_0$ is analogous to the retrosynthesis problem (Corey, 1989; Segler et al., 2018). We employ this approach of training $P_B$ against a different objective than $P_F$ to palliate to a design challenge of the MDP, but a similar strategy could also be employed to fold additional preferences over different synthesis routes leading to the same terminal state into the system, for example to take into account the synthesis costs of a particular path.

### 3.3 Scaling of the reactant space to large number of building blocks

**Estimating the size of the state space** We leverage the properties of GFlowNets (Bengio et al., 2021) to estimate the size of the state space induced by our action space. Specifically, noting that GFlowNets learn $\log Z = \log \sum_x R(x)$, we train a model with $R = 1$ for all terminal states to estimate their total count. We do so for different numbers of building blocks and different maximum trajectory lengths ($L$), and find that SynFlowNet using $L = 3$ and 10k building blocks matches the size of the Enamine REAL space (Enamine), and that the size of the space quickly increases with the number of building blocks. We use our full set of 105 reactions. Note how reaction-constrained models considerably limit the exploration of the chemical space, with a fragments GFlowNet exploring a space $\sim$10 orders of magnitude larger.

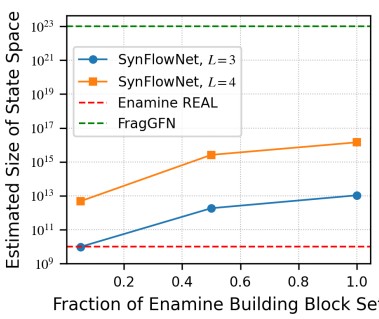

Figure 2: **Estimated size of state spaces.** The full building block set contains 221,181 molecules. $L$ is the maximum trajectory length in the GFlowNet.

**Scaling method** To ensure synthetic accessibility of all samples, our model is inherently constrained by the initial set of available building blocks (BBs). To cover a large chemical space, it is thus crucial to use an extensive BB collection. The model must demonstrate scalability to accommodate larger sets of BBs, both in terms of training efficiency and overall performance. To do so, we follow Dulac-Arnold et al. (2015); Gottipati et al. (2020a) and change the representation of the BBs and their selection mechanism, as shown in Figure 1B. Instead of the weight matrix of the mapping from hidden units to logits associated with BBs to be randomly initialized, it is *fixed* to be the matrix of binary Morgan fingerprints (Rogers & Hahn, 2010).

## 4 Results

Our results support a number of claims: (i) a reaction-based MDP greatly improves the synthesisability of generated molecules (Sec. 4.1), (ii) employing GFlowNets enables much more diverse molecule sampling over RL (Sec. 4.5), (iii) learning a backward policy results in higher-reward molecules and enables finding retrosynthetic pathways for molecules belonging to our state-space (Sec. 4.4), and (iv) our method of sampling molecules based on chemical fingerprints allows for efficient scaling to large chemical spaces (Sec. 4.5). Finally, we show that this approach also allows to improve results for particular programs by curating the set of building blocks based on target-specific experimental data (Sec 4.6). Unless otherwise specified, all experiments below use a backward policy trained with maximum likelihood, and Morgan fingerprint embeddings for our action space, with a library of 10,000 Enamine building blocks and 105 reaction templates.

### 4.1 Reaction-based MDP design

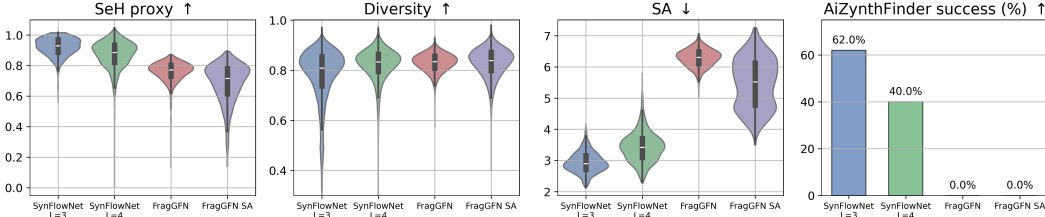

Figure 3: **Comparison across MDPs.** We evaluate four GFlowNet models: SynFlowNet is trained with an action space of chemical reactions and maximum trajectory lengths (L) of 3 and 4, FragGFN and FragGFN SA are trained with an action space of fragments, however the latter also optimises for synthetic accessibility (SA), besides the sEH binding proxy reward. SynFlowNet molecules are achieving higher binding scores and better synthesizability.

We compare GFlowNets using two action spaces – fragments vs. reactions – and report reward, diversity and synthesizability of the generated samples. We retrain the GFlowNet model proposed by

Bengio et al. (2021) with fragments derived from our building blocks set. This ensures that we use similar chemical spaces and that we get as close as possible to a fair comparison between the two MDPs. In the fragment space, it is common practice to optimise for synthetic accessibility scores to improve synthesizability from the model. We therefore train two versions of the FragGFN model: one using the sEH binding proxy as reward function, and another that optimizes for both sEH binding and synthetic accessibility (SA) score. SynFlowNet was only trained with the sEH proxy as reward.

The results in Fig. 3 show that our MDP design achieves great improvement in terms of synthesizability, while preserving high rewards and diversity. While one might expect the more expressive FragGFN model to result in higher rewards, we hypothesize that the larger exploration space in the fragments environment (see Figure 2) hinders the model's efficiency. We see that within the reaction environment, MDP design choices can further influence sample quality: a maximum trajectory length of 3 achieves better SA scores and AiZynthFinder (Genheden et al., 2020) retrosynthesis success (62%) compared to a maximum trajectory length of 4 (40%). Note that synthesisability metrics are correlated with molecule size (Skoraczyński et al., 2023), and that molecules assembled from longer synthesis routes are naturally larger (see Table A.3). Both FragGFN models score 0% with AiZynthFinder. Diversity, measured as average Tanimoto distances between molecular fingerprints, is preserved from a fragments to a reaction environment, albeit less so for shorter trajectories.

## 4.2 GFlowNets as samplers of chemical space

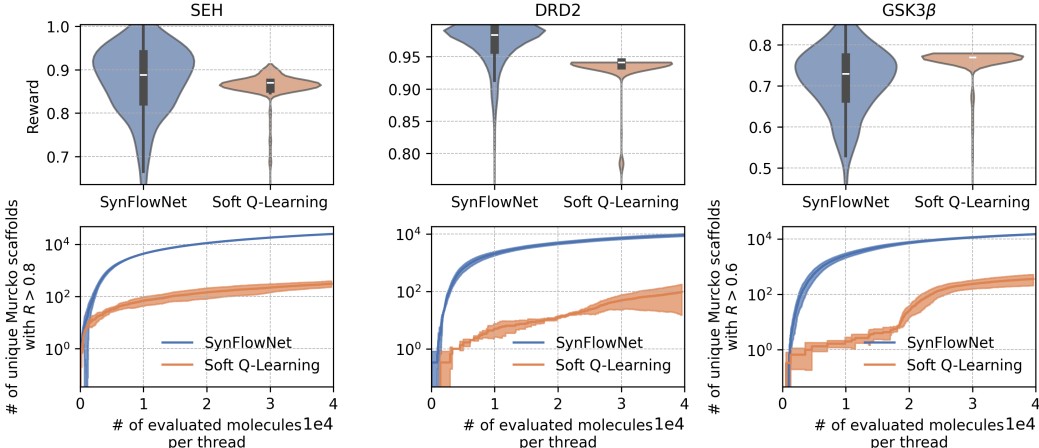

Figure 4: **Comparison between GFlowNet and RL.** SynFlowNet (GFlowNet with synthesis actions) discovers more modes compared to entropy-regularised RL trained on the same state & action space.

While the works of Gottipati et al. (2020b) and Horwood & Noutahi (2020) are closest to our method, where RL is paired with a synthesis action space, their code is not publicly available. We reproduce their setting by pairing an RL algorithm with our MDP. We train our model with a soft Q-learning (Haarnoja et al., 2017) objective and compare to SynFlowNet on three different reward functions: sEH, GSK3$\beta$ and DRD2 (see Appendix A.1).

In Fig. 4, we show the reward distribution of generated samples after equal numbers of training steps. We notice that soft Q-learning collapses to sampling a narrower distribution of high-reward molecules for all targets. This is also shown in the number of Bemis-Murcko scaffolds counted for molecules with rewards above a certain threshold, where soft Q-learning quickly collapses.

## 4.3 Further comparison to baselines

We continue by comparing SynFlowNet to strong baselines from the literature. First, REINVENT uses a policy-gradient method to tune an RNN pre-trained to generate SMILES strings (Blaschke et al., 2020; Loeffler et al., 2024). REINVENT has been shown to outperform many models in terms of sample efficiency (Gao et al., 2022a) and proposing realistic 3D molecules upon docking (Ciepliński et al., 2023). Second, SyntheMol shares a similar MDP design with SynFlowNet, but is a search-based method using Monte Carlo Tree Search (Swanson et al., 2024). Contrary to our model,

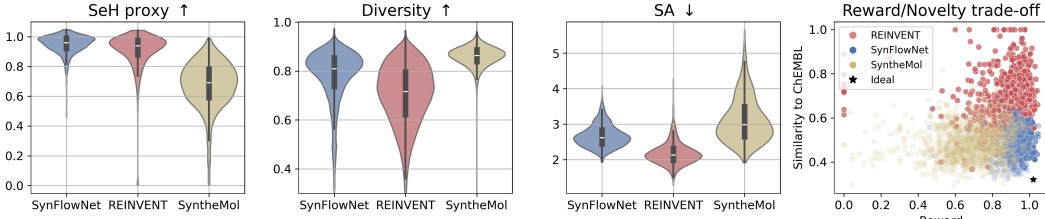

Figure 5: **SynFlowNet is competitive against other popular models from the literature.** Syn-FlowNet achieves a good balance between reward optimisation, diversity, synthesizability and novelty (assessed by maximum similarity to ChEMBL molecules). REINVENT stays close to its pretraining distribution, harming the novelty of the proposed molecules. SynFlowNet is closest to ideal.

synthesis pathways in SyntheMol contain one reaction, and its estimated state space has 30 billion molecules, reached from the Enamine building block set and 13 reactions (Swanson et al., 2024).

The comparison is summarized in Figure 5. SynFlowNet achieves comparable rewards to REINVENT, and better sample diversity. Although REINVENT does not use explicit knowledge of synthesizability in its generation process, it implicitly optimizes for it by being a likelihood model overfit on a curated ChEMBL dataset (Zdrazil et al., 2023). This is reflected in our last sub-plot, where we look at the maximum similarity to ChEMBL molecules, as a proxy for novelty. This proxy is strong enough, as ChEMBL contains one of the largest collection of diverse drug-like molecules to date (Zdrazil et al., 2023). While REINVENT seems to be competitive in terms of synthesizability and reward, it generates molecules with poor novelty, staying close to its pretraining distribution. Notably, SynFlowNet achieves high novelty for high-scoring molecules, even when trained using ChEMBL-derived building blocks (see App. Fig. A.5). Overall, SyntheMol scores comparably to SynFlowNet, however SynFlowNet achieves a better reward/diversity trade-off and SynFlowNet's capabilities are extended to explore larger state spaces (see Fig. 2). We repeat the comparison on other targets, and again find that SynFlowNet outperforms SyntheMol (see App. B.1). Further experiments showcasing SynFlowNet's sample efficiency and performance when optimizing for direct docking score calculations, are discussed in Appendix B.1. We again discover that SynFlowNet is competitive to other models and that it improves sample efficiency over the fragments GFlowNet.

### 4.3.1 REDISCOVERY TASKS

We investigate whether SynFlowNet can discover *known* synthesizable actives by training it with rediscovery rewards (Huang et al., 2021). The results are shown in Table 1, where we report top-$k$ similarity and SA scores. When running the SpaceLight software (BioSolveIt, 2024) to determine whether the two molecules are present in the REAL space (i.e. reachable through our set of building blocks), we found that aripiprazole is present and that celecoxib is not, which explains the two rates.

Table 1: **SynFlowNet performance on rediscovery tasks.** We report mean and standard deviation of top-10 discovered molecules from running three models with different seeds.

| Task | Reward ($\uparrow$) | SA ($\downarrow$) |
|---|---|---|
| Aripiprazole rediscovery | $0.90 \pm 0.00$ | $2.19 \pm 0.00$ |
| Celecoxib rediscovery | $0.48 \pm 0.01$ | $2.47 \pm 0.04$ |

### 4.4 IMPROVED MDP CONSISTENCY THROUGH TRAINED BACKWARD POLICY

We evaluate the effectiveness of our proposed training schemes for the backward policy $P_B$ by measuring (1) whether $P_B$ can ensure that backward-constructed trajectories reliably find a trajectory back to the initial state $s_0$ and (2) whether it brings any benefit to the forward policy. In Table 2 we compare a fixed, uniform backward policy to three versions of a parameterized backward policy: a free policy, updated w.r.t. the trajectory balance loss, (see Eq. 1), a policy trained with maximum likelihood on the forward-generated trajectories and a policy that is allowed to explore the backward action space and trained with REINFORCE to find paths leading back to $s_0$. We observe that both the maximum likelihood and REINFORCE policies succeed in ensuring that backward flow is not lost outside of the MDP, as they manage to construct trajectories that start from terminal states sampled from $P_F$ all the way to $s_0$. We refer to such routes as *solved routes (train)* in Table 2, as the terminal

states have been visited by the GFlowNet during training. We also test the ability of the trained policies to retrieve synthesis routes for molecules which have not been visited during training ($test$). These were obtained from a random sampler using our set of reactions and building blocks. While we see small differences in the number of high-reward modes discovered for different backward policies, we see that a free policy consistently fails to be competitive with the rest. More ablation studies in App. A.6 show that an entropy-regularized REINFORCE policy excels at discovering high reward modes. Overall, the maximum likelihood and REINFORCE policies prove effective in guiding $P_F$ to high reward modes, while ensuring MDP consistency, which is crucial for sampling from the Boltzmann distribution defined by the reward function $R$ (Bengio et al., 2023).

| $P_B$ policy | % of solved routes (train) | % of solved routes (test) | # of high reward modes from $P_F$ |
|---|---|---|---|
| Uniform | $11.0 \pm 3.7\%$ | $11.0 \pm 4.1\%$ | $47,515.0 \pm 11,264.7$ |
| Free | $67.3 \pm 3.7\%$ | $1.0 \pm 0.8\%$ | $6,754.3 \pm 4,980.1$ |
| MaxLikelihood | $99.3 \pm 0.5\%$ | $32.3 \pm 7.3\%$ | $37,708.2 \pm 13,992.6$ |
| REINFORCE | $100.0 \pm 0.0\%$ | $44.3 \pm 2.6\%$ | $55,387.6 \pm 28,886.3$ |

Table 2: **Effect of different training paradigms for $P_B$.** Training the GFN $P_B$ ensures that backward-constructed trajectories belong to our MDP and can marginally improve $P_F$. % of solved routes refers to the ability of $P_B$ to retrieve synthesis routes for on-policy (train) and off-policy (test) molecules reachable through our state space. Test molecules have not been seen during training. # of high reward ($R > 0.9$) modes from $P_F$ is reported out of $\sim 500,000$ samples seen during training.

## 4.5 SCALING TO THE ENTIRE ENAMINE SET

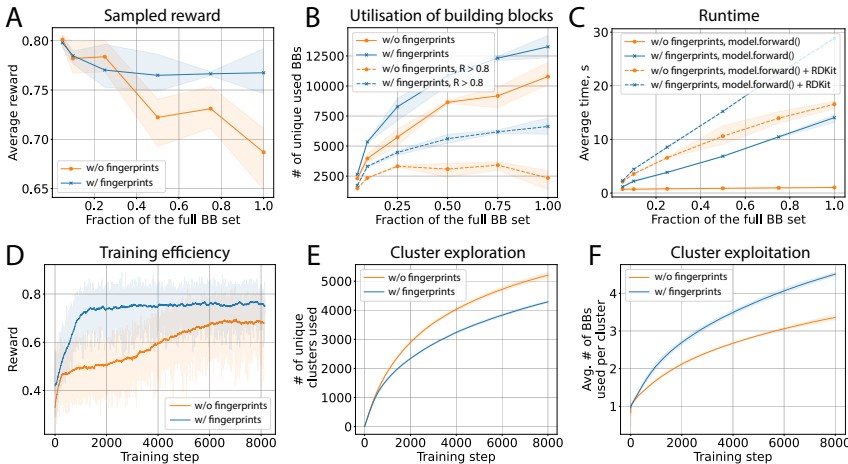

Figure 6: **Ablation over action-selection mechanism.** Average reward of samples (A), utilization of building blocks (B), and training times (C) for both models on different subsets of building blocks. Embeddings enable more efficient learning (D), as they help the agent navigate the set of available building blocks and tend to exploit more relevant BB clusters (F) rather than explore new clusters (E). For each experiment, three models trained with different random seeds were used.

In this section, we study the ability of SynFlowNet to scale to large sets of building blocks using the method proposed in Section 3.3 as opposed to using a learnable embedding for each building block. First, we compare the performance of our models using the softmax action-selection mechanism and the fingerprints-based softmax on the full Enamine set of $221,181$ building blocks and randomly sampled subsets containing $75\%$, $50\%$, $25\%$, $10\%$, and $3\%$ of the full set. As shown in Figure 6A, increasing the number of building blocks while using the same number of training steps negatively affects the reward for the model without fingerprints. However, the model with fingerprints consistently outperforms the former and makes the reward degradation effect almost negligible. Next, we study whether models are able to use the blocks they are given. As shown in Figures 6B, the model with fingerprints consistently uses more unique building blocks than the baseline model. Notably, the

baseline model is only able to use a small subset of building blocks to produce high-quality samples ($R > 0.8$). In contrast, fingerprints enable the use of a 5-fold larger set of blocks for the same quality level, thus significantly surpassing the baseline samples in terms of diversity. Finally, we measure the average time of the forward pass and average time of forward pass with auxiliary RDKit operations like template matching in Figure 6C.

As shown in Figure 6D, fingerprints enable more efficient learning on a large building block set. Indeed, instead of memorising all available BBs, the model learns to navigate in the space of embeddings and to maximise dot product with the relevant building blocks. To illustrate this, we clustered all available building blocks based on Tanimoto similarity. Next, we studied how both models explore new building blocks within and across these clusters. As shown in Figures 6E-F, the baseline model tends to explore more clusters while discovering low-reward molecules, thus becoming less efficient. The model with fingerprints, on the contrary, is able to focus on clusters that maximise the reward, leveraging the structure of the chemical space embedded in fingerprints.

### 4.6 GUIDING SYNFLOWNET WITH EXPERIMENTAL FRAGMENT SCREENS

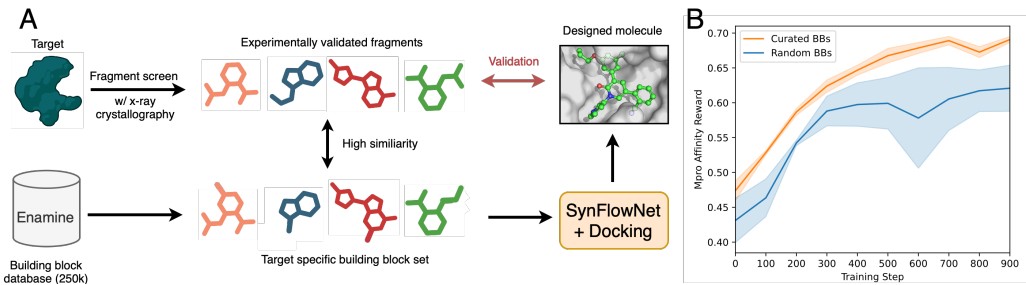

Figure 7: **Effect of curating the building block set.** (A) We adapt SynFlowNet's building block library based on experimentally validated fragments for a given target. (B) The curated set improves Mpro rewards over random building blocks.

One advantage of our framework is that building block sets can be used to specialize the model for a particular target. Fragment-based Drug Discovery (FBDD) (Thomas et al., 2019) is a major strategy to increase the efficiency of drug discovery campaigns. In FBDD, instead of screening large chemical spaces, a relatively low number of small compounds (called fragments) are screened and experimentally validated to bind, and are then linked or merged *in silico* to make full molecules.

We hypothesise that experimental data from x-ray fragment screens (Murray & Rees, 2009) can guide and enhance SynFlowNet's capabilities. As a proof-of-concept, we focus on the SARS-CoV-2 main protease (Mpro), leveraging strucural data from the COVID Moonshot project (Boby et al., 2023). We compile a building block set from Enamine with high similarity to fragments confirmed by x-ray crystallography (see Fig. 7A). We find that biasing the building block library towards fragments with known protein-ligand complementarity increases the reward over randomly selected molecules (see Fig. 7B). Further details on methods and results are given in App. Section B.4.

### 5 CONCLUSION

In this work, we discuss the application of GFlowNets to *de novo* molecular design paired with forward synthesis. We demonstrate that an action space of chemical reactions is an effective way of enforcing synthesisability, and that pairing it with GFlowNets excels in terms of diversity. Comparisons to state-of-the-art baselines emphasised that SynFlowNet explores novel regions of the chemical space. We also proposed a novel paradigm for training the backward policy in the GFlowNet and in doing so we improved and validated the correctness of our MDP design. Furthermore, we studied the building block exploration and exploitation mechanisms of SynFlowNet, showing efficient scaling to using hundreds of thousands of building blocks. Finally, as a proof of concept for the adaptability of SynFlowNet to real drug discovery programs, we showed that the framework can be specialised to target-specific molecule generation by making use of experimental fragment screens.

ACKNOWLEDGEMENTS

We thank Enamine for providing the building blocks and for useful conversations. We thank Daniel Cutting, Austin Tripp and Jeff Guo for discussions regarding baselines. We thank Doo-Hyun Kwon (Recursion) for running the SpaceLight software. Miruna Cretu is supported by the SynTech CDT at the University of Cambridge and part of the work was completed during an internship at Valence Labs.

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

# A EXTENDED METHODS

## A.1 REWARD FUNCTIONS

We train SynFlowNet for a number of reward functions/targets. In some cases (e.g. when benchmarking against the Fragment-based GFlowNet), we also perform multiple objective optimisation with SA score.

**sEH** Our main reward function is defined as the normalized negative binding energy as predicted by a pretrained proxy model, available from Bengio et al. (2021) and trained on molecules docked with AutoDockVina (Trott & Olson, 2010) for the sEH (soluble epoxide hydrolase) protein target, a well studied protein which plays part in respiratory and heart disease (Imig & Hammock, 2009). The proxy model, which utilizes the weights from Bengio et al. (2021), was trained using a message-passing neural network (MPNN) (Gilmer et al., 2017) that processes atom graphs as input. Details of the model architecture are available in Bengio et al. (2021). It was trained on a dataset of 300,000 randomly generated molecules, achieving a test mean squared error (MSE) of 0.6. Note that the reward scale in our results differs from the original GFlowNet publication, with rewards adjusted by a factor of 1/8 in our analysis.

**GSK3$\beta$ and DRD2** We also employ two oracle functions from the PMO Gao et al. (2022a) benchmark, which provide machine learning proxies trained fit to experimental data to predict the bioactivities against their corresponding disease targets. The two targets we use here are GSK3$\beta$ Li et al. (2018) and dopamine receptor D2 (DRD2) (Olivecrona et al., 2017a).

**Easy adoption to other targets using GPU-accelerated Vina docking** Finally, we wish for users to rapidly be able to adapt SynFlowNet to learn binding for their target of interest without having to retrain a new proxy or relying on slow docking simulations. We accomplish this using the new GPU-accelerated Vina-GPU 2.1 docking algorithm (Tang et al., 2023; Alhossary et al., 2015). For our experiments, we use the `PDB:6W63` (Mesecar et al., 2020) structure for the SARS-CoV-2 main protease (Mpro) target, `PDB:2XJX` (Murray et al., 2010) for Heat Shock Protein 90 (HSP90) target and `PDB:8AZR` for KRAS (Kim et al., 2023). Receptors are prepared for docking with `prepare_receptor4.py` and the center of the docking is defined as the center of mass for the ligand with a size of 25 Å in accordance with previous work (Buttenschoen et al., 2024). In order to prevent the model from optimising for large molecules, we add a reward penalty of $-0.4$ for molecules with a number of heavy atoms larger than that of the reference ligand plus an allowance of 8 additional heavy atoms. The Vina scores are scaled between 0 and 1 according to:

$$R = \frac{affinities + reward\_scale\_min}{reward\_scale\_min + reward\_scale\_max} - 1, \tag{4}$$

where $reward\_scale\_min$ and $reward\_scale\_max$ are tunable and set to `-1.0` and respectively `-10.0` by default.

## A.2 DATA

We use commercially available building blocks (BBs) from Enamine, which are small fragments of molecules prepared in bulk to be readily synthesised into candidate molecules. Reaction templates are obtained from two publicly available template libraries (Button et al., 2019; Hartenfeller M, 2012). After preprocessing, we obtain a total of 105 templates: 13 uni- and 92 bi-molecular reactions respectively (see App. A.2.1). We also use 12 Enamine REAL reactions in a small number of experiments (see Section 4.6 and App. A.2.1).

### A.2.1 CHEMICAL REACTIONS

**Reaction pre-processing and action masking** Reaction templates pre-processing was necessary to ensure that the templates could be run backwards. All templates containing wildcards (*) in the reactant SMARTS were duplicated with replacements for all atoms that the wildcard substituted for.

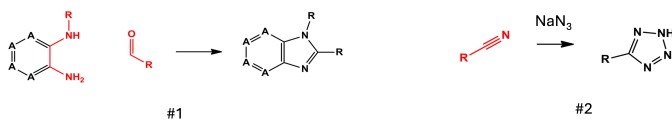

Figure A.1: **Example of reaction templates** (Hartenfeller M, 2012). The templates act as rules, which match any molecule that left side (before the arrow) as a subgraph. The matching part is then transformed into the right hand side of the rule. "R" represents any group, "A" represents an aromatic atom. Note the implicit reagent used in reaction #2.

When running experiments with a REINFORCE backward policy (which requires to sample trajectories backward on-policy), we enforced additional masking that forward reactions are sampled if and only if they are reversible (i.e. when applying the template backward on the product, the same reactants are obtained as the ones utilised by the template in the forward direction). Note that reaction templates can be *uni-molecular* (employing one reactant, Fig. A.1 #2) or *bi-molecular* (employing two reactants, Fig. A.1 #1).

**Enamine REAL Space Reactions**   For the case study in Section 4.6, we used reactions from Enamine to stay close to an experimental setup where the molecules generated would be readily purchasable from Enamine. Enamine assembled a molecular space called the REAL space, which is a vast catalogue of 48B purchasable compounds[0]. We use a subset of Enamine reactions available from Swanson et al. (2024) which produces 93.9% of the REAL space[1].

### A.3   MODEL AND TRAINING

A graph neural network based on a graph transformer architecture (Yun et al., 2019) is used to parameterize the forward and backward policies. The model's action space is defined using separate MLPs for each action type (see Figure 1). Our model is trained in an online fashion, meaning that it learns exclusively from trajectories sampled from the GFlowNet policy, without relying on an external dataset of trajectories or a set of target molecules. Note however that this framework is compatible with offline training, which makes use of such datasets as starting point for exploring the molecular space.

**SynFlowNet Training**   We adapt the framework from Bengio et al. (2021) to train a GFlowNet sampler over a space of synthesisable molecules, which are assembled from an action space of chemical reactions and reactants. A graph neural network with a graph transformer architecture (Yun et al., 2019) is used to produce a state-conditional distribution over the actions. A state is represented as a molecular graph in which nodes contain atom features. Edge attributes are bond type and the indices of the atoms which are its attachment points. This representation is augmented with a fully-connected virtual node, which is an embedding of the conditional encoding of the desired sampling temperature, obtained using an MLP. The sampling temperature is controlled by a temperature parameter $\beta$, which also plays a role in reward modulation, allowing for exponential scaling of the rewards (by making rewards received during training equal to $R^\beta$). We experimented with sampling $\beta$ from multiple distributions, and use a constant distribution in the reported results in this paper. We used a thermometer encoding of the temperature (Buckman et al., 2018).

The model is trained using the trajectory balance objective (Malkin et al., 2022) and thus is parameterized by forward and backward action distributions $P_F$ and $P_B$ and an estimation of the partition function $Z = \sum_{\tau \in \mathcal{T}} F(\tau)$.

For the state space estimation experiment, we used $R = 1$, $10\,000$ training steps and varying trajectory lengths. For the rest of the SynFlowNet experiments with SeH proxy as reward, we used $\beta = 32$, 5000 training steps and varying trajectory lengths (see main text). For training with the DRD2 and GSK3$\beta$ targets as reward, we use a maximum trajectory length of 4 and a reward exponent $\beta = 16$. For the backward policy training, see App. A.4. The rest of the hyperparameters are fixed and are

---

[0]https://enamine.net

[1]Note: this dataset contains a single tri-molecular reaction which has been removed here for simplicity

presented in Table A.1. For the multi-objective optimization experiments in Sections 4.1 and 4.2, we multiplied the different rewards.

**FragGFN training**   We obtained fragments and their attachment points from Enamine building blocks by following the protocol provided by Jin et al. (Jin et al., 2019). The model trained with the sEH reward was trained with the default implementation in Yoshua Bengio (2024), and an optimised reward exponent $\beta = 64$. The model optimising for synthetic accessibility, as well as sEH binding, was trained with a reward obtained by multiplying the two scores.

**Soft Q-learning training**   We implemented a version of Soft Q-learning (SQL) (Haarnoja et al., 2017), an energy-based policy learning method, that operates on SynFlowNet's MDP. We optimised the training procedure by performing both manual and grid searches across several values of entropy regularisation parameter $\alpha$ and reward scaling parameter $\beta$. For $\beta$, contrarily to GFlowNets, since the method does not only learn a policy but tries to estimate the Q-values of each actions directly, we found that using large values of reward scaling such as $\beta = 64$, which are common for GFlowNets, would destabilise the algorithm and had to be lowered to $\beta = 4$ or $\beta = 2$. For $\alpha$, we tried several values that would strike the best tradeoff between allowing the model to find high-performing molecules while maximising diversity and avoiding to collapse the agent's distribution on only a few modes, with our best model using $\alpha = 0.01$.

**REINVENT training**   We benchmarked our approach against REINVENT4 (Loeffler et al., 2024)[2]. Following the setup described in their methodology, we fine-tuned the REINVENT prior model using reinforcement learning. We change little else other than the reward function used to train the model. The training was conducted with the default batch size of 100 over 3,000 training steps, resulting in a total of 300,000 oracle calls, which is consistent with the number of oracle calls used in SynFlowNet experiments.

**SyntheMol generation**   We run SyntheMol's (Swanson et al., 2024) Monte Carlo Tree Search (MCTS) algorithm using their standard sets of $139\,493$ building blocks and 13 chemical reactions from the Enamine REAL space, but replace the provided bioactivity prediction models with the reward functions used in this study (Sec. A.1). We use the publicly available code repository[3] and perform $50\,000$ rollouts. Other hyperparameters were kept at default settings, including a maximum of 1 reaction and exploration parameter $c = 10$. Before the search, building block scores were pre-computed using the target reward functions. For the final selection, we sample 1000 molecules randomly from all returned molecules.

| Hyperparameters | Values |
| --- | --- |
| Batch size | 64 |
| Number of GNN layers | 4 |
| GNN node embedding size | 128 |
| Graph transformer heads | 2 |
| Learning rate ($P_F$) | $10^{-4}$ |
| Learning rate ($P_B$) | $10^{-4}$ |
| Learning rate ($Z$) | $10^{-3}$ |

Table A.1: Hyperparameters used in our SynFlowNet training pipelines.

## A.4   BACKWARD POLICY TRAINING ALGORITHMS

For the maximum likelihood backward policy, at each training iteration, $P_B$ is updated to minimize the maximum likelihood loss over trajectories generated in that batch from $P_F$. Similarly for a REINFORCE $P_B$, we train $P_F$ as above, but maintain a replay buffer. To train $P_B$, we sample terminal states from the buffer and sample trajectories backwards from $P_B$, which are used in a REINFORCE update. To improve training stability, we also used trajectories generated from $P_F$ (in a 1:1 ratio) to update $P_B$.

---

[2]We used the code available at `https://github.com/MolecularAI/REINVENT4`
[3]`https://github.com/swansonk14/SyntheMol`

Contrary to the rest of the models (see App. A.3), the backward policy models were trained for 8000 steps, with reward exponent $\beta = 64$ and max_len $= 5$. The results in Section 4.4 are reported for a REINFORCE loss with an entropy multiplier term $\alpha = 1.0$.

---

**Algorithm 1** Training of Maximum Likelihood Backward Policy for GFlowNets

---

1: **Initialize** the forward policy $P_F$, backward policy $P_B$, and $Z_\theta$.
2: **repeat**
3:     Sample a batch of trajectories $\{\tau^{(n)}\}_{n=1}^N$ from $P_F$.
4:     Update $P_F$ and $Z_\theta$ to minimize $\mathcal{L}_{TB}$ using $\{\tau^{(n)}\}_{n=1}^N$.
5:     Update $P_B$ to minimize $\mathcal{L}_B$ over $\{\tau^{(n)}\}_{n=1}^N$.
6: **until** convergence

---

**Algorithm 2** Training of REINFORCE Backward Policy for GFlowNets

---

1: **Initialize** the replay buffer $\mathcal{B}$, forward policy $P_F$, backward policy $P_B$, and $Z_\theta$.
2: **repeat**
3:     Sample a batch of trajectories $\{\tau_F^{(n)}\}_{n=1}^N$ from $P_F$.
4:     Update $\mathcal{B} \leftarrow \mathcal{B} \cup \{\tau_F^{(k)}\}_{n=1}^N$.
5:     Sample $k$ random trajectories from $\mathcal{B}$ and extract their final states $s_f$ to sample backward trajectories $\{\tau_B\}_{k=1}^K$ from $P_B$.
6:     Update $P_F$ and $Z_\theta$ to minimize $\mathcal{L}_{TB}$ using $\{\tau_F^{(n)}\}_{n=1}^N$.
7:     Update $P_B$ to minimize $J_B$ over $\{\tau_B^{(k)}\} \cup \{\tau_F^{(n)}\}$.
8: **until** convergence

---

## A.5 SYNFLOWNET SCALING

Here, we provide technical details of the experiments described in Section 4.5. For each building block subset, we trained 3 models using fingerprints, and 3 models without fingerprints, which amounts to 36 models in total. All models had exactly the same hyper-parameter configuration with SeH proxy as reward, 8000 training steps, batch size 8 and $\beta = 32$. Other parameters were the same as in Table **??**

For evaluation, we sampled 5000 trajectories with each model and considered only trajectories with valid final molecules. In Figure 6C, we report average runtime per batch over the training process. `model.forward()` corresponds to the average time required to perform the matrix operations (PyTorch back-end, single GPU NVIDIA H100) during the forward pass of the model and aims to highlight the performance overhead introduced by the dot product with molecular fingerprints. `model.forward() + RDKit` additionally accounts for the costly template matching operations ran on CPU after the model's forward pass (to shortlist the set of allowed building blocks for the chosen reaction). These values demonstrate the effect of the increased building block set. To characterize the structural diversity of the molecules generated by different models, we additionally provide the average pairwise Tanimoto dissimilarity of the molecules sampled by different models (using different building block subsets) in Figure A.2.

For Figures 6D-F, the entire building block set was used. To compute clusters of building blocks for Figures 6E-F, we used BitBIRCH (Jung et al., 2024), a recent adaptation of Balanced Iterative Reducing and Clustering using Hierarchies (BIRCH) algorithm (Zhang et al., 1996), recently proposed for efficient clustering of large molecular libraries.

## B EXTENDED RESULTS

### B.1 COMPARISON TO BASELINES

Table A.3 contains metrics for all the models and baselines we have run. We additionally show results from training SynFlowNet with a reward measuring binding affinity to KRAS using GPU-accelerated VINA docking (Tang et al., 2023), and comparison to baselines. Figure A.4 contains results for

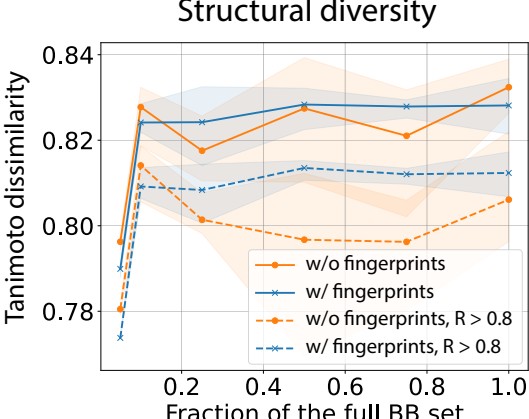

Figure A.2: Diversity of the molecules produced by the baseline and fingerprint models trained with different building block subsets.

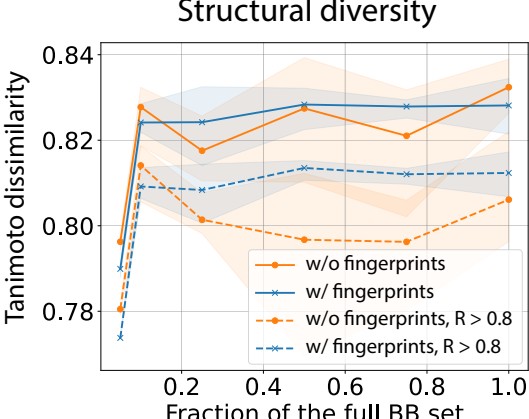

Figure A.3: **Example of synthesis pathway generated by SynFlowNet.** The reward here is the sEH binding affinity proxy. Right shows the Vina docked pose and molecule metrics. More examples are shown in Figure A.7.

molecules generated with SyntheMol for the DRD2 target, compared to SynFlowNet. SynFlowNet is superior in finding high-reward molecules. We tried various hyperparameters for SyntheMol to optimise for the GSK3$\beta$ target and were unsuccessful – no nonzero rewards were found.

| Method | sEH proxy (↑) | Diversity (↑) | SA (↓) | AiZynth (↑) | QED (↑) | Mol. weight (↓) | ChEMBL Similarity (↓) |
|---|---|---|---|---|---|---|---|
| REINVENT | $0.91 \pm 0.01$ | $0.68 \pm 0.02$ | $2.19 \pm 0.02$ | 0.95 | $0.57 \pm 0.04$ | $429.78 \pm 23.02$ | $0.64 \pm 0.02$ |
| FragGFN | $0.77 \pm 0.01$ | $0.83 \pm 0.01$ | $6.28 \pm 0.02$ | 0.00 | $0.30 \pm 0.01$ | $724.62 \pm 32.39$ | $0.24 \pm 0.09$ |
| FragGFN SA | $0.70 \pm 0.01$ | $0.83 \pm 0.01$ | $5.45 \pm 0.05$ | 0.00 | $0.29 \pm 0.01$ | $683.31 \pm 59.92$ | $0.21 \pm 0.01$ |
| SyntheMol | $0.64 \pm 0.01$ | $0.86 \pm 0.01$ | $3.08 \pm 0.01$ | 0.82 | $0.63 \pm 0.01$ | $412.24 \pm 0.98$ | $0.49 \pm 0.01$ |
| Soft Q-Learning | $0.80 \pm 0.07$ | $0.42 \pm 0.04$ | $2.63 \pm 0.39$ | 0.96 | $0.39 \pm 0.02$ | $408.02 \pm 12.76$ | $0.52 \pm 0.02$ |
| SynFlowNet ($L=3$) | $0.92 \pm 0.01$ | $0.79 \pm 0.01$ | $2.92 \pm 0.10$ | 0.65 | $0.59 \pm 0.02$ | $365.23 \pm 2.42$ | $0.43 \pm 0.01$ |
| SynFlowNet SA ($L=3$) | $0.94 \pm 0.01$ | $0.75 \pm 0.02$ | $2.67 \pm 0.03$ | 0.93 | $0.68 \pm 0.01$ | $358.27 \pm 3.52$ | $0.48 \pm 0.01$ |
| SynFlowNet ChEMBL ($L=3$) | $0.91 \pm 0.02$ | $0.80 \pm 0.01$ | $2.68 \pm 0.18$ | 0.67 | $0.68 \pm 0.01$ | $342.67 \pm 8.00$ | $0.49 \pm 0.01$ |
| SynFlowNet ($L=4$) | $0.88 \pm 0.01$ | $0.82 \pm 0.01$ | $3.54 \pm 0.03$ | 0.40 | $0.27 \pm 0.01$ | $557.49 \pm 8.60$ | $0.38 \pm 0.01$ |
| SynFlowNet QED | $0.86 \pm 0.03$ | $0.81 \pm 0.03$ | $4.02 \pm 0.26$ | 0.55 | $0.74 \pm 0.04$ | $398.50 \pm 8.84$ | $0.38 \pm 0.01$ |

Table A.2: **Comparison to baselines.** Results obtained by averaging over 1000 random molecules sampled from the trained models. Standard errors obtained from training using 3 seeds. SynFlowNet ChEMBL refers to a model trained with building blocks derived from ChEMBL molecules. Due to high computational cost, AiZynthFinder scores are computed over 100 random samples.

| Method | KRAS Vina (↑) | Diversity (↑) | SA (↓) | QED (↑) | Mol. weight | ChEMBL Similarity (↓) |
|---|---|---|---|---|---|---|
| REINVENT | $0.76 \pm 0.03$ | $0.85 \pm 0.01$ | $2.51 \pm 0.05$ | $0.53 \pm 0.03$ | $359.52 \pm 6.51$ | $0.57 \pm 0.01$ |
| FragGFN | $0.86 \pm 0.06$ | $0.62 \pm 0.11$ | $3.54 \pm 0.02$ | $0.43 \pm 0.03$ | $348.89 \pm 34.42$ | $0.42 \pm 0.04$ |
| Soft Q-Learning | $0.66 \pm 0.01$ | $0.88 \pm 0.02$ | $3.10 \pm 0.05$ | $0.61 \pm 0.01$ | $370.97 \pm 12.76$ | $0.44 \pm 0.02$ |
| SynFlowNet ($L=3$) | $0.73 \pm 0.09$ | $0.87 \pm 0.01$ | $3.05 \pm 0.10$ | $0.56 \pm 0.08$ | $389.91 \pm 23.27$ | $0.42 \pm 0.01$ |

Table A.3: **KRAS binding optimization with GPU-accelerated Vina docking.** Results obtained by averaging over 1000 random molecules sampled from the trained models. Standard errors obtained from training using 3 seeds. Vina scores are scaled between 0 and 1 using Eq. 4.

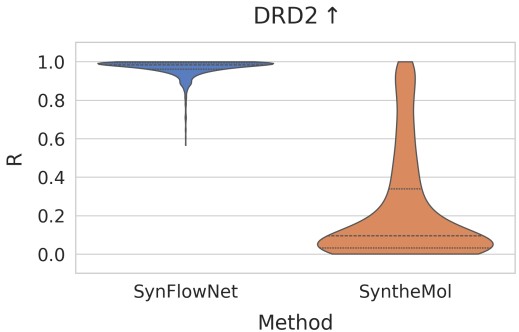

Figure A.4: Comparison between SynFlowNet and SyntheMol with DRD2 as reward. SyntheMol struggles to find high-reward molecules. We also run SyntheMol with GSK3$\beta$ and were not able to optimise it. Results for SynFlowNet with both DRD2 and GSK3$\beta$ are reported in Figure 4.

We further compare SynFlowNet's performance to SynNet's (Gao et al., 2022b), which models the generation of synthetic trees containing multi-step synthesis pathways as a Markov decision process. The approach differs from SynFlowNet and Gottipati et al. (2020a); Horwood & Noutahi (2020) in that it relies on a trained model optimized using a genetic algorithm, instead learning the target distribution solely from the reward function, based on an RL objective. In Table A.4 we report top-100 performance on the optimization tasks performed in SynNet.

Table A.4: **Comparison to SynNet.** We report average of top-100 scores. Results for SynNet are reported from Gao et al. (2022b).

| Method | QED | GSK3$\beta$ | DRD2 | JNK3 |
|---|---|---|---|---|
| SynNet | 0.947 | 0.815 | 0.998 | 0.719 |
| SynFlowNet | 0.947 | 0.862 | 0.999 | 0.710 |

**Novelty**    For a more strict evaluation of the novelty of SynFlowNet-proposed designs, we investigate whether SynFlowNet can still generate low-similarity molecules to ChEMBL when relying on building blocks derived from ChEMBL actives. In this way, we reach comparable vocabularies between REINVENT's prior and SynFlowNet's action space. For this, we select 70 000 random ChEMBL molecules from Zdrazil et al. (2023) and run AiZynthFinder retrosynthesis (Genheden et al., 2020) to decompose the molecules into building blocks. This results in 8527 unique building blocks which were used to train SynFlowNet. We report comparative results from this experiment in Figure A.5 and Table A.3 and note that SynFlowNet preserves high novelty compared to REINVENT, which emphasizes the advantages of the framework.

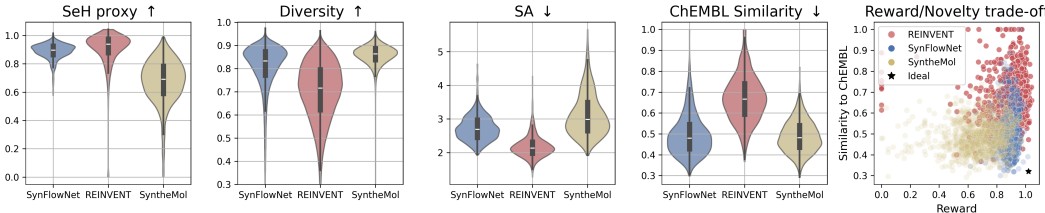

Figure A.5: SynFlowNet trained with ChEMBL-derived building blocks remains competitive against other popular models from the literature. SynFlowNet achieves the best novelty/reward and diversity/reward trade-offs.

### B.1.1   SAMPLE EFFICIENCY

Following Gao et al. (2022a), we benchmark SynFlowNet against other generative models for molecular design in terms of sample efficiency. We focus on the two reward functions (DRD2 and

GSK3$\beta$) that are included in the benchmark and report the area under the curve (AUC) of the top-10 average performance versus oracle calls. Following Gao et al. (2022a), we limit the maximum number of oracle calls to $10\,000$ and report the results in Table A.5.

Table A.5: **Sample efficiency results.** We report mean and standard deviation of the AUC of top-10 average scores versus the number of oracle calls from 5 independent runs. Results for all other methods are reported from Gao et al. (2022a).

| Task | Method | AUC |
|------|--------|-----|
| GSK3$\beta$ | REINVENT | $0.865 \pm 0.043$ |
| | SynNet | $0.789 \pm 0.032$ |
| | FragGFN | $0.651 \pm 0.026$ |
| | SynFlowNet | $0.691 \pm 0.034$ |
| DRD2 | REINVENT | $0.943 \pm 0.005$ |
| | SynNet | $0.969 \pm 0.004$ |
| | FragGFN | $0.590 \pm 0.070$ |
| | SynFlowNet | $0.885 \pm 0.027$ |

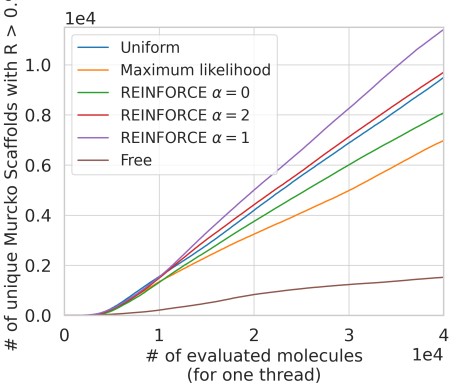 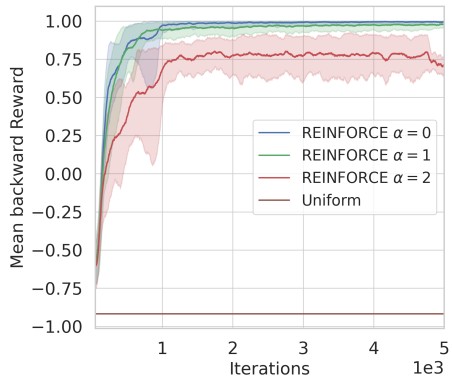

Figure A.6: (LHS) Effects of different parameterizations of the backward policy. For REINFORCE, ($\alpha$) represents the multiplier to the entropy term. (RHS) REINFORCE is trained on-policy and the mean rewards of the sampled backward trajectories are plotted. Note that the backward reward is 1 if the backward-constructed trajectory ends in $s_0$ and -1 otherwise. A baseline for Uniform policy is shown, obtained by sampling 100 random trajectories backward from terminal objects in the DAG. Results averaged over 8 seeds.

## B.2 OUTLOOK

While SynFlowNet shows promising results for generating high-quality synthesizable molecules, there remain challenges to be addressed in future work. First, SynFlowNet does not explore non-linear synthetic pathways, i.e. considering intermediates in the DAG as second reactants. At the moment the choice of second reactants is limited to the building blocks library. SynFlowNet also does not handle reactions with more than 2 reactants. We also emphasize that reaction selectivity is not accounted for, and that including reaction feasibility in the design process would strengthen real-world synthesizability results. Another inherent limitation of the reaction templates employed in this work is the lack of consideration of reaction conditions and stereochemistry. We envision that the backward policy in SynFlowNet can be an innovative way of accounting for additional constraints in the MDP design, and our preliminary results in Section 3.2 on training the backward policy with a separate reward from the forward policy support this. One could further bake in constraints such as synthesis cost, reaction yield and selectivity in the backward reward $R_B$. Additionally, a more careful curation of the reaction set is encouraged in future work, which would enhance the coverage of the chemical space via diverse transformations: cyclizations, side-chain extensions, functional group

interconversions etc. Lastly, the rediscovery rate of SynFlowNet is not perfect, meaning that certain regions of the chemical space may remain inaccessible to the model under the current datasets.

## B.3 Example Trajectories

Figure A.7 shows examples of molecules and synthesis pathways generated from SynFlowNet.

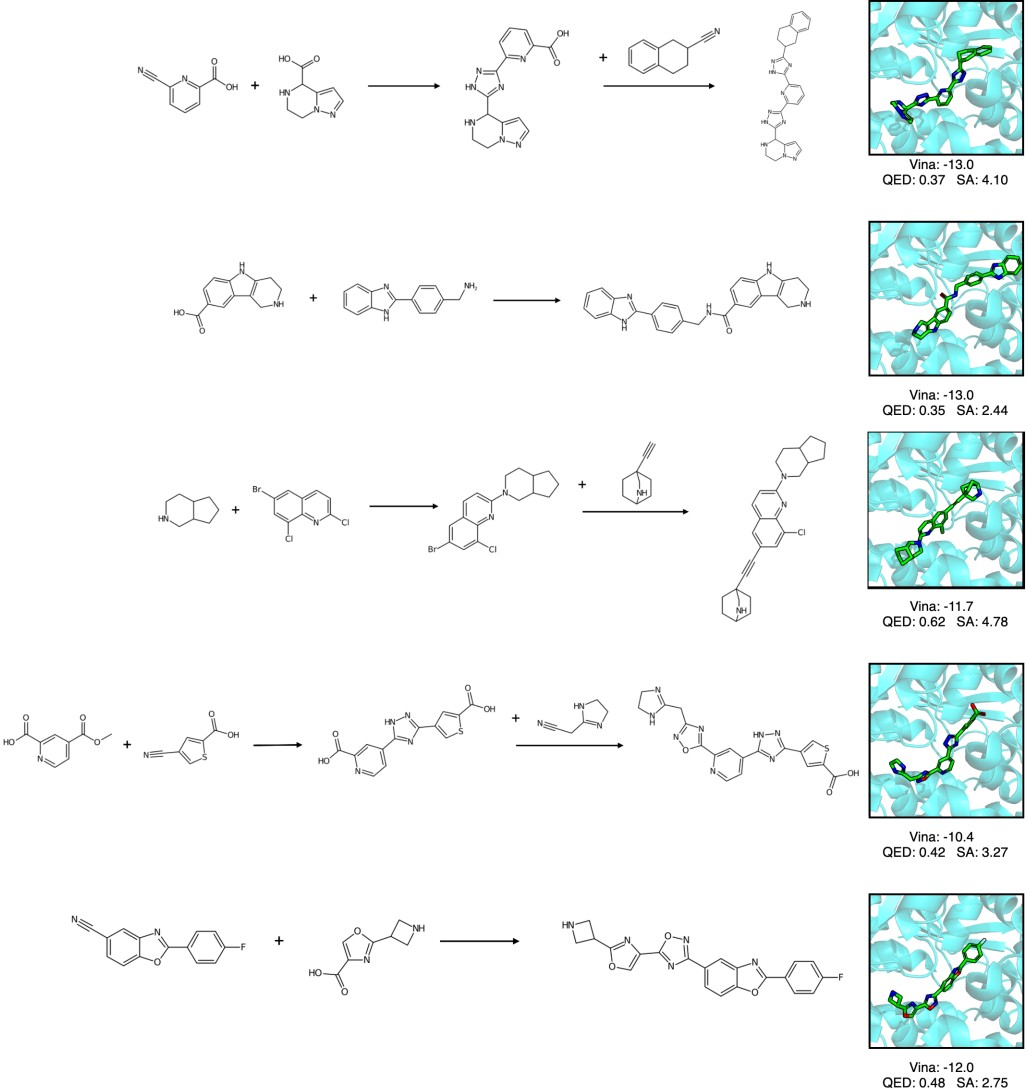

Figure A.7: SynFlowNet-generated molecules and synthesis pathways for SeH.

### B.4 CURATED BUILDING BLOCK SETS GIVEN TARGET DATA

Here we provide further details on the methods and rationale for the fragment-based building block curation strategy outlined in 4.6. To enable high molecule purchaseability in a real world drug discovery campaign, we use the Enamine reactions in all these experiments. GPU-accelerated Vina docking was used for reward computation (see App. A.1).

**Experimental fragment extraction pipeline**   Fragment-based drug discovery (FBDD) simplifies the search space by focusing on smaller molecular fragments rather than designing or screening full molecules, enabling a more efficient exploration of potential binding interactions (Murray & Rees, 2009; Thomas et al., 2019). To identify molecules bound within the same pocket across different protein structures, we performed a sequence-based search using MMSeqs2 (Steinegger & Söding, 2017) across the Protein Data Bank (PDB). We retained only hits with a sequence identity of 90% or greater and an alignment overlap exceeding 80% to ensure high similarity to the reference target. The resulting PDB structures were structurally aligned[4] according to the ligand-bound chain with the reference structure. Any ligand with at least one atom within 2 Å of the reference ligand in the binding site was selected for further analysis.

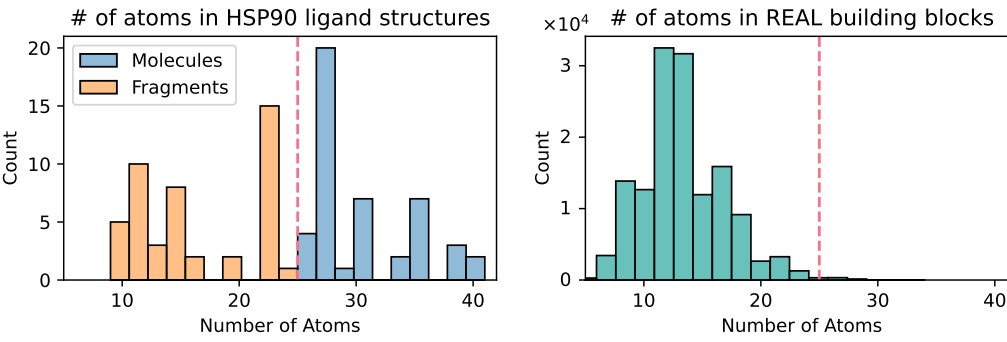

Figure A.8: Curation of HSP90 fragments and building blocks. Red line shows threshold at which we denote a molecule a 'fragment' and was choosen to max the upper limit of the typical BB size.

**Selection of small molecule fragments**   To isolate small molecular fragments suitable for FBDD, we filtered out ligands containing more than 25 atoms, as this atom count aligns with typical building block molecule sizes. This is done for 2 reasons; (i) we can pretend that we are in a drug discovery campaign where a fragment screen has just been conducted and there is no leaked information from 'full' molecules and (ii) this threshold aligns well with the typical Enamine building block in terms of size (Figure A.10).

**Curation of Enamine Building Blocks**   For every experimentally validated fragment, we selected the top 100 closest building block molecules from the Enamine library based on molecular similarity. After removing duplicate SMILES entries, this process usually resulted in thousands of curated building blocks for each target.

**Example fragment screens**   We study two targets. The first is Heat Shock Protein 90 (HSP90), for which there are a large number of ligand bound structures, mostly thanks to FBDD campaigns conducted in industry (Murray et al., 2010; Woodhead et al., 2010). The second is the SARS-CoV-2 Main Protease (Mpro), for which there is a large amount of structural data and in particular fragments from a fragment screen performed by the COVID Moonshot open-science initiative (Boby et al., 2023). In the case of HSP90, using PDB entry 2XJX (Woodhead et al., 2010) as the reference structure, our ligand extraction pipeline identified 92 molecules. Filtering for small molecular fragments of fewer

---

[4]We use the `superimpose_homologs` function from biotite: `www.biotite-python.org`

than 25 atoms further refined this set for analysis. The methodology was repeated for Mpro using PDB entry 7GAW, where we extracted 138 molecules.

Examples of molecules designed for Mpro using our curated building block set are shown in Figure A.9.

**Alignment of SynFlowNet designs with experiments** Murray et al. (2010) performed a fragment screen of HSP90 and identified two distinct lead classes of binding modes: (i) an aminopyrimidine class that formed hydrogen bonds with an asparagine residue and several conserved water molecules within the pocket and (ii) a phenolic class that primarily binds through water-mediated hydrogen bonds networks. These complicated water-dependent interaction networks are extremely challenging to model computationally and is a clear limitation of Vina docking (which does not take into account waters). Figure A.10 shows that by guiding SynFlowNet in this way, we can design molecules and poses that fit with these experimentally validated interactions.

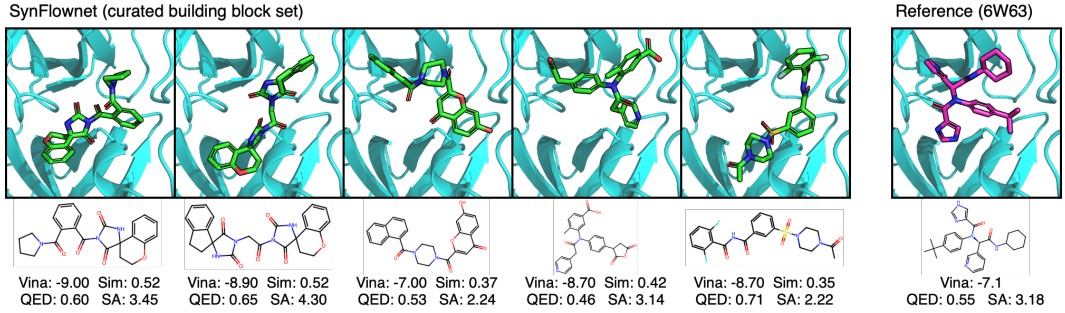

Figure A.9: **Example molecules generated by SynFlowNet for the Mpro case study.** Green molecules are those generated by SynFlowNet. Magenta shows the molecule from the reference structure. 'Sim' is the Tanimoto similarity between a designed molecule and the reference structure.

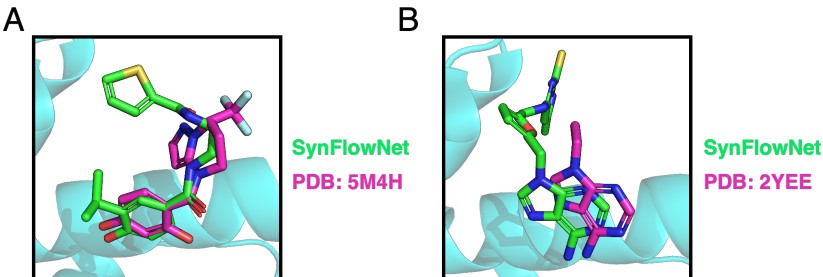

Figure A.10: **SynFlowNet designs molecules that align with real world fragment experiments.** Murray et al. (2010) performed a fragment screen for the HSP90 target and identified two classes of binding modes for the target: those built around (A) phenolic compounds (e.g. PDB:5M4H) and (B) aminopyrimidine compounds (e.g. PDB:2YEE). SynFlowNet trained on a target-curated building block set was able to consistently design compounds based on these experimentally validated binding modes, meaning we can be more certain that they are likely to bind well in reality. Green shows SynFlowNet-designed molecule and magenta are fragments from x-ray crystallography experiments.

