# OpenReview forum: "SynFlowNet: Design of Diverse and Novel Molecules with Synthesis Constraints"
_ICLR.cc/2025/Conference — ICLR 2025 Spotlight_

### Official Review · Reviewer_AADH · 2024-10-22

**Soundness:** 2
**Presentation:** 3
**Contribution:** 3
**Rating:** 8
**Confidence:** 4

**Summary:**

This paper presents SynFlowNet, a practical approach for drug design that incorporates reaction-based Markov Decision Processes (MDPs) into a Generative Flow Network (GFlowNet) to generate diverse molecules with guaranteed synthetic pathways.
Notably, authors propose a novel backward policy training strategy to ensure the presence of backward trajectories.
Furthermore, the authors investigate the potential of SynFlowNet to be integrated with wet experiments such as fragment-based crystallography.

**Strengths:**

1. Novel Backward Policy of GFlowNet training: SynFlowNet proposes a novel learning strategy for backward policy to ensure the directed acyclic graph and facilitate synthesizable molecules.
2. Connection to Wet-Lab Experiments: The case study in Section 4.6 provides a clear demonstration of how to connect wet-lab experiments and *in silico* drug design, highlighting the potential real-world applicability of the method.
3. Comprehensive Ablation Studies: The paper provides ablation studies for many design choices.

**Weaknesses:**

**Overall Comment**

I acknowledge the novelty of this work in successfully adapting the GFlowNet framework for synthesizable molecular design, but there are several critical aspects of the current manuscript that need to be addressed.
I think this manuscript may be suitable for acceptance after revision, and I’m positive about changing the scores during the rebuttal period.

*\* The issues are sorted in the order they appeared in the manuscript.*

**Issue 1: Lack of novelty / appropriate citations for the major contribution.** Page 2, Line 103, Contribution 3

One major concern is that the manuscript does not contain appropriate citations for this major contribution.
To use 200k building blocks, the authors used action embedding [1], which has already been employed in existing synthesis-oriented molecular generative models [2-4].
In particular, Koziarski et al. [4] introduced action embedding into GFlowNet and investigated its effectiveness in large action spaces.
While the authors use a different fingerprint representation (Morgan fingerprints instead of MACCS fingerprints), Gottipati et al. [2] have already evaluated various representations, including both fingerprints.
Authors should highlight the novel differences from existing approaches or provide appropriate citations.

**Issue 2: Lack of experimental details.** Section 4.1 and Section 4.5
- In Section 4.1, what is the rewarding strategy used for multi-objective optimization (FragGFN SA, SynFlowNet SA, SynFlowNet QED)? The possible reward structures could be Multi-objective GFlowNet (MOGFN) [5] or multiplication-based rewards [6].
- In Section 4.5, I couldn't find the experimental details. The related section is on Page 19, Section A.5, but it does not provide crucial information about this experiment. I suggest to include experimental details and mention it like Section 4.6.
  - Reward Function: What reward function was used in the ablation studies? Is it sEH?
  - Average Times: Clarify what *average times* refers to–e.g., runtime per batch or runtime per molecule.
  - For Figure 7 D-F, entire building block set is used?

**Issue 3: Fairness of evaluation metric.** Page 8, Figure 6

The improvement in the reward/diversity trade-off is impressive, but I have questions about the metric, **Reward/Novelty trade-off**.
Regarding the Reward/Novelty trade-off, it is possible that the improvement in novelty is due to the use of the Enamine building blocks rather than the effectiveness of well-structured generative model architecture.
In this case, the REINVENT's high similarity to ChEMBL would be greatly reduced by simply retraining it on Enamine REAL or ZINC.

For a fair comparison in terms of novelty, the comparison should be performed on the same exploration space:
- Train REINVENT on Enamine REAL.
- Use ChEMBL-like building blocks, such as blocks obtained from the retrosynthetic decomposition [3] of ChEMBL molecules.

**Issue 4: Limited comparative study.** Sections 4.2 and 4.3

The comparison between the proposed method and baselines was performed only on proxy models.
I recognize that proxy models such as sEH [7] are widely used to evaluate the effectiveness of optimization strategies in the optimization community [8], so I think this is a minor issue.
However, I have concerns about the appropriateness of evaluating and comparing the performance of discovering novel molecules based on proxy models trained on a restricted and known chemical space.

**Issue 5: Restricted scalability to a large building block set.** section 4.5

I definitely agree with the argument for employing an extensive building block (BB) set to explore the vast chemical space.
I believe that it can mitigate the reward-diversity trade-off by expanding the sample space.
However, I concern that the current experimental results do not seem to provide sufficient evidence about this argument and the scalability of SynFlowNet.

The authors demonstrated that a larger BB set yielded a higher selection of unique BBs (Figure 7B), but this has a negative impact on the average reward (Figure 7A) and speed (Figure 7C).
Consequently, the current findings seem to emphasize the negative effects of the usage of extensive BB sets on the quality and throughput for generation.
To justify the significance of utilization of unique BBs, I suggest including structural diversity metrics for generated samples, such as the number of unique Bemis-Murcko scaffolds or the average pairwise Tanimoto distance.

---
**Reference**
1. Dulac-Arnold, Gabriel, et al. "Deep reinforcement learning in large discrete action spaces." arXiv preprint arXiv:1512.07679 (2015).
2. Gottipati, Sai Krishna, et al. "Learning to navigate the synthetically accessible chemical space using reinforcement learning." International conference on machine learning. PMLR, 2020.
3. Seo, Seonghwan, Jaechang Lim, and Woo Youn Kim. "Molecular generative model via retrosynthetically prepared chemical building block assembly." Advanced Science 10.8 (2023): 2206674.
4. Koziarski, Michał, et al. "RGFN: Synthesizable Molecular Generation Using GFlowNets." ICML'24 Workshop ML for Life and Material Science: From Theory to Industry Applications (2024).
5. Jain, Moksh, et al. "Multi-objective gflownets." International conference on machine learning. PMLR, 2023.
6. Lee, Seul, Jaehyeong Jo, and Sung Ju Hwang. "Exploring chemical space with score-based out-of-distribution generation." International Conference on Machine Learning. PMLR, 2023.
7. Bengio, Emmanuel, et al. "Flow network based generative models for non-iterative diverse candidate generation." Advances in Neural Information Processing Systems 34 (2021): 27381-27394.
8. Gao, Wenhao, et al. "Sample efficiency matters: a benchmark for practical molecular optimization." Advances in neural information processing systems 35 (2022): 21342-21357.

**Questions:**

1. **Page 2, Figure 1B:**
    - As I understand it, the building block selection probability depends on not only the state embedding but also the selected reaction. (Page 4 Line 213). However, the Figure 1B indicates that the selection probability is calculated only from the state information $s_k$. I suggest revising the figure and including the series of equations for the neural network to clarify this process.
    - Also, I was wondering if dot product operation between Morgan fingerprint (binary vector) and state embedding can satisfy the GFlowNet Objective. When the 4-bit fingerprints for building blocks $b_1$, $b_2$, and $b_3$ are $[1,0,0,0]$, $[0,1,0,0]$, and $[1,1,0,0]$, respectively, the probability of $b_3$ can be derived from those of $b_1$ and $b_2$:
      $$\log P(b_3|s) = \frac{\log P(b_1|s) + \log P(b_2|s)}{\sqrt{2}}$$
      I think that this relationship can introduce bias in the forward transition probability.
3. **Page 7, Footnote:**
Is it typo?: '… which are different to the **reactants** used by SynFlowNet …'. As I know, AiZynthFinder allows user to use different reactants. I think 'reactants' should be replaced to 'reactions'.
4. **Page 9, Figure 7C:**
I suggest changing the title "Performance" to "Runtime" or a similar term.
5. **Page 19, Table A.1 (Hyperparameters)**:
Could you clarify the meaning of *max_nodes*? In the standard GFlowNet implementation, it refers to the maximum number of heavy atoms in molecules, but the generated molecules illustrated in the paper have more than 9 heavy atoms.

---

> ### Author Response · Authors · 2024-11-20
> **Author response (1/2)**
>
> We thank the reviewer for providing valuable feedback and insights. We are addressing their questions one by one in our response below.
>
> > **Issue 1: Lack of novelty / appropriate citations for the major contribution**
> >
> > the manuscript does not contain appropriate citations for this major contribution [action embeddings]
>
> We provided references to the existing methods that use similar action space embeddings in Section 3.3.
>
> > **Issue 2: Lack of experimental details.**
> >
> > what is the rewarding strategy used for multi-objective optimization (FragGFN SA, SynFlowNet SA, SynFlowNet QED)?
>
> The strategy that we used what the multiplication of rewards, which has now been noted in the manuscipt, in Section A.3.
>
> > In Section 4.5, I couldn't find the experimental details. I suggest to include experimental details and mention it like Section 4.6.
>
> We have added a detailed description of scaling experiments in Section A.5, including what reward function was used, clarification with regards to average time, and building blocks set used.
>
> > **Issue 3: Fairness of evaluation metric**
> >
> >  it is possible that the improvement in novelty is due to the use of the Enamine building blocks rather than the effectiveness of well-structured generative model architecture
>
> We thank the reviewer for the suggestion to train SynFlowNet with ChEMBL-derived building blocks. We have conducted this experiment by decomposing randomly-sampled ChEMBL molecules into building blocks using AiZynthFinder retrosynthesis. The results (presented in Section B.1, Table A.2 and Figure A.5) show that SynFlowNet designs preserved their lower similarity to ChEMBL, while still performing well on reward and diversity. This suggests that although relying on a ChEMBL vocabulary, the model has the capability to extrapolate to new regions of the chemical space, as opposed to models relying on likelihood-based priors for optimisation like REINVENT. Indeed, the scope of the original experiment was to show that REINVENT, although it generates highly synthesizable and high-reward molecules, it will not manage to extrapolate to novel chemical space, and that SynFlowNet represents a more unbiased solution on the sampling problem.
>
> | Method                       | sEH proxy (↑)       | Diversity (↑)       | SA (↓)            | AiZynth (↑) | QED (↑)          | Mol. weight (↓) | ChEMBL Similarity (↓) |
> |------------------------------|---------------------|---------------------|-------------------|-------------|------------------|-----------------|-----------------------|
> | REINVENT                    | 0.91 ± 0.01        | 0.68 ± 0.02        | 2.19 ± 0.02       | 0.95        | 0.57 ± 0.04     | 429.78 ± 23.02 | 0.64 ± 0.02          |
> | SynFlowNet (L = 3)      | 0.92 ± 0.01   | 0.79 ± 0.01    | 2.92 ± 0.10   | 0.65    | 0.59 ± 0.02 | 365.23 ± 2.42 | 0.43 ± 0.01      |
> | **SynFlowNet ChEMBL (L = 3)**   | **0.91 ± 0.02**        | **0.80 ± 0.01**        | **2.68 ± 0.18**       | **0.67**        | **0.68 ± 0.03**     | **342.67 ± 8.00**  | **0.49 ± 0.01**          |
>
> > **Issue 4: Limited comparative study**
> >
> > I have concerns about the appropriateness of evaluating and comparing the performance of discovering novel molecules based on proxy models trained on a restricted and known chemical space.
>
> Following the reviewer's comment, we have added comparisons between models trained to optimise for a KRAS binding affinity reward, calculated using a GPU-accelerated version of AutoDock Vina [1]. Such an oracle enables exploration beyond the restricted and known chemical space associated with other proxy models. Results are presented in Table A.3 in Appendix and support the claims that SynFlowNet is generating molecules with the best reward/diversity/novelty trade-off.

---

> ### Author Response · Authors · 2024-11-20
> **Author response (2/2)**
>
> > **Issue 5: Restricted scalability to a large building block set.**
> >
> > I suggest including structural diversity metrics for generated samples, such as the number of unique Bemis-Murcko scaffolds or the average pairwise Tanimoto distance
>
> The purpose of Figure 7 ([edit]: now Figure 6) is two-fold: to show how SynFlowNet scales with increasing number of building blocks and to show the effect the utilisation of Morgan fingerprints for action embeddings. We show that the effect is positive and that the model with fingerprints exploits more within the same clusters, meaning that it is able to leverage the known structure of the chemical space embedded by Morgan fingerprints. Validating that the number of unique utilised building blocks increases with larger action space is an important check that the model does not collapse and has the capacity to scale and explore increasingly larger action spaces. We also clarify that the results in Figure 7 ([edit]: now Figure 6) are for runs following the same number of training steps, therefore the decrease in reward for larger action spaces is expected. In fact, we managed to reach the same rewards as for the smaller action spaces when increasing the computational budget.
>
> Following the reviewer's suggestion, we include the average pairwise Tanimoto dissimilarity as a structural diversity metric in Figure A.2, and see that the model consistently samples diverse molecules.
>
> > Question 1: I was wondering if dot product operation between Morgan fingerprint (binary vector) and state embedding can satisfy the GFlowNet Objective
>
> That is an interesting question to consider. Indeed, classical binary fingerprints could induce representational limitations on the model. In the example that the reviewer is mentioning, involving fingerprints [1,0,0,0], [0,1,0,0] and [1,1,0,0], if the first two features are deemed beneficial according to the current state, the third molecule, represented by [1,1,0,0] would always be preferred over the first two and its sampling likelihood would be a function of the other two.
>
> We investigated a bit further how frequent this case might be. In general, it is a combinatorial problem to verify whether any of the building blocks can be obtained by simply summing (logical OR) the binary representation of some combination of other building blocks, and the computation time grows exponentially with the building block size and the group size (pairs, triples, etc.). We ran this test for the set of 10,000 building blocks which used in the paper, and found that using 1024-dimensional Morgan Fingerprints, there was not any occurrence where a building block's representation could be obtained by summing those of a pair of other building blocks. For these reasons, we believe that this potential representational limitation does not have a practical effect on our models.
>
> > Questions 2-4:
>
> Thank you for pointing out the typos and confusing figure title, as well as the misspecified _max_nodes_. These have been corrected in the updated manuscript.
>
> ---
> References
>
> [1] Tang, Shidi et al., "Vina-GPU 2.1: towards further optimizing docking speed and precision of AutoDock Vina and its derivatives", 2023: https://www.biorxiv.org/content/10.1101/2023.11.04.565429v1

---

> > ### Comment · Reviewer_AADH · 2024-11-22
> > **Official Comment by Reviewer AADH**
> >
> > Sorry for delayed comments for the response.
> >
> > I found that authors provide sufficient revisions for my response. I'll increase the score to 6 or 8.
> > I'll decide the score after reviewing other impressive modifications for the reviews from other reviewers within 24 hours.
> >
> > There are additional minor comments for the revised manuscript:
> > - Page 26 Ln 1361: "Figure ?? shows that by guiding SynFlowNet": Please modify this error.
> > - Page 21 Table A.3: "Vina scores are scaled between 0 and 1.": Please include the scaling factor.
> > - Page 21 Table A.3: "Mol. weight (↓)": The active ligand of KRAS is C23H27N7OS and the molecular weight is about 450. The range of appropriate MW varies from pocket to pocket, so small is better may not be the right.

---

> ### Author Response · Authors · 2024-11-22
> **Thank you for the positive feedback**
>
> Thank you for pointing these out, and for taking the time to review our modifications. The additional formatting issues / comments have now been addressed in the updated manuscript.
>
> We look forward to hearing from the reviewer and welcome further suggestions!

---

> ### Comment · Reviewer_AADH · 2024-11-22
> **Official Comment by Reviewer AADH**
>
> Thank you for immediate updates for my comments.
>
> Now I think the paper is sufficient to be accepted, so I'll increase the score from 5 to 8.
>
> ---
> Regardless of the score, I also have an additional question, which would be the last.
> A lot of time has already passed in the review period, and I've been able to find enough meaning in all experiments.
> This is just a question from my curiosity, so **you do not have to make a rebuttal or response to this question** by spending your valuable time.
> If you have time, please feel free to answer them.
>
> In Section 4.6, the authors construct a target-specific building block set using fragment screening results from the COVID Moonshot project.
> However, in my best knowledge, the Moonshot project includes large-scale fragment screening results (common fragments), so it is unclear whether the selected building blocks are truly target-specific.
> Is there any additional progress to select target-specific fragments among the common fragments?
> If not, what factor do you think caused the improvement?

---

> > ### Author Response · Authors · 2024-11-24
> > **Thank you for increasing the score!**
> >
> > Thank you for increasing the score and for the question!
> >
> > Yes, additional progress is involved in the screening to ensure that the fragments used are target specific. Indeed, fragment screening experiments typically start with ~1000 fragments (which are common across all experimental screens and not target specific). However, after extensive binding affinity experiments (and determination of 3D structures via X-ray crystallography) a typical screening campaign yields 50-100 fragments that are confirmed to bind to the target, and these are the fragments that we used to construct our curated building blocks set. They were extracted from the Protein Data Bank using the pipeline explained in Appendix Section B.4, paragraphs *Experimental fragment extraction pipeline* and *Selection of small molecule fragments*. The selection ensured that we actually extracted Mpro ligands that were also outside of the COVID Moonshot campaign, as multiple other entries were done outside of the campaign. In paragraph *Example fragment screens* we explain that this pipeline resulted in 138 fragments that we used further to select Enamine building blocks. In the paper we state that the choice of Mpro was motivated by the large amount of structural data obtained thanks to the COVID Moonshot project, but we should indeed clarify that we did not rely directly on the large-scale COVID Moonshot screening results, but used the pipeline described in the appendix.
> >
> > Please let us know if this clarifies the question and thanks for the interest!

---

> > > ### Comment · Reviewer_AADH · 2024-11-24
> > > **Thank for your additional response.**
> > >
> > > Thank you for additional response.
> > >
> > > I was just wondering if the fragment screening results meant the entire collected data for the all input fragments(~1000) or selected one (50-100). This response is enough to address my curious, and thank you for answering!

---

### Official Review · Reviewer_cHNB · 2024-10-29

**Soundness:** 2
**Presentation:** 2
**Contribution:** 2
**Rating:** 6
**Confidence:** 4

**Summary:**

The paper presents SynFlowNet, a model designed to improve the generation of novel drug-like molecules while ensuring their synthetic accessibility. SynFlowNet constructs molecules sequentially using commercially available reactants and documented chemical reactions, effectively constraining the generation process within a synthesizable chemical space. The model integrates a forward synthesis constraint, evaluates synthetic accessibility through external retrosynthesis tools, and optimizes diversity and reward (e.g., binding affinity) without relying on reinforcement learning, which tends to sample fewer, high-reward molecules. SynFlowNet introduces a backward policy for navigating reaction trajectories, improving synthetic feasibility and sampling accuracy. By utilizing molecular fingerprints and scalable building blocks, SynFlowNet can explore vast chemical spaces efficiently, making it adaptable for specific drug targets when provided with experimental fragment screening data, such as in SARS-CoV-2 protease inhibitors.

**Strengths:**

- Focus on Synthetic Accessibility: SynFlowNet adopts a framework that effectively addresses a key limitation in generative models for drug design by ensuring that generated molecules are synthetically feasible, which is crucial for practical drug development applications.
- Diverse and Optimized Molecule Sampling: Leveraging GFlowNets, SynFlowNet provides a balanced sampling process that maximizes both reward and structural diversity, effectively mitigating the typical mode collapse encountered in reinforcement learning-based models.
- Validation through Synthetic Accessibility Metrics: In silico experiments demonstrate SynFlowNet’s advantage over fragment-based approaches, particularly in terms of synthesizability, underscoring the practical benefits of synthesis-based molecule generation for drug discovery.

**Weaknesses:**

Limited Testing on Optimization Tasks: While SynFlowNet showcases improvements in SeH proxy, its optimization abilities are not fully tested against established molecular optimization benchmarks. Incorporating PMO benchmark tasks and directly comparing results with other generative baselines, such as those in Gao et al. (2022) [1], could provide a more comprehensive evaluation of SynFlowNet’s optimization performance.

### Reference
[1] Gao, Wenhao, et al. "Sample efficiency matters: a benchmark for practical molecular optimization." Advances in neural information processing systems 35 (2022): 21342-21357.

**Questions:**

NA

---

> ### Author Response · Authors · 2024-11-20
> **Author response (1/1)**
>
> We thank the reviewer for the provided feedback and we address the points raised below.
>
> > Incorporating PMO benchmark tasks and directly comparing results with other generative baselines, such as those in Gao et al. (2022) [1], could provide a more comprehensive evaluation of SynFlowNet’s optimization performance.
>
> We have now incorporated a study of SynFlowNet's sample efficiency, based on the suggested benchmark by Gao et al [1]. The results and discussion are presented in Section B.1.1 in Appendix and show that SynFlowNet is more sample efficient than its fragments analogue. We would also like to emphasize that generative models based exclusively on sampling from a reward function, such as SynFlowNet, are mostly designed for relatively inexpensive oracle functions, such as medium-accuracy docking simulations, or high-throughput screening platforms which can produce hundreds of thousands of readouts each weeks or months, therefore be believe that the reported sample efficiency of SynFlowNet is satisfactory.
>
> Please let us know if there are any further questions/suggestions!
>
> ---
> References
>
> [1] Gao, Wenhao, et al. "Sample efficiency matters: a benchmark for practical molecular optimization." Advances in neural information processing systems 35 (2022): 21342-21357.

---

> > ### Comment · Reviewer_cHNB · 2024-11-25
> >
> > Thank you for conducting the additional experiment to test the sample efficiency. The results enhance the comprehensiveness of the evaluation and provide a more thorough perspective on the proposed method. I would like to maintain my initial assessment unchanged.

---

### Official Review · Reviewer_6JHQ · 2024-11-03

**Soundness:** 3
**Presentation:** 3
**Contribution:** 4
**Rating:** 8
**Confidence:** 4

**Summary:**

The authors proposed a molecular generative model that explicitly considers the synthesizability of generated molecules. Specifically, they formulated a molecular design problem into a Markov decision process (MDP) that successively adds a purchasable building block (BB) with a selected reaction to the previous state. A GFlowNet has been adopted to tackle the MDP. Namely, SynFlowNet shows better performance in improving the diversity and high reward modes simultaneously, whereas conventional optimization approaches like reinforcement learning only focus on the highest reward modes. To handle more BBs, they are represented with binary Morgan fingerprints, and the probability of sampling a particular BB is computed as a function of its fingerprint and the embedding vector of the current state. In addition,  new backward policies have been tested to ensure that the backward trajectories can return to existing building blocks. They also performed a case study that leverages experimental data for more efficient drug design as a practical application. Overall, the manuscript is well-written, but several issues need to be addressed.

**Strengths:**

1. The use of the GFlowNet improves both diversity and high reward modes compared to other optimization methods such as reinforcement learning.
2. The proposed backward policy ensured that backward-constructed trajectories belong to the GFlowNet MDP.
3. Various benchmark and ablation studies were performed to ensure the strategies of the proposed framework.

**Weaknesses:**

1. The conclusions of some experiments are arguable (see the questions below), which need to be justfied probably with additional experiments for clarity.
2. Increasing the number of BBs slows down training linearly that may cause a computational burden when dealing with a large amount of BBs.

**Questions:**

1. In Table 1, while the two backward policies, the Maxlikelihood and REINFORCE, show relatively high ratios of the solved routes in both the training and test sets, their number of high reward modes is similar to that of the Uniform Policy. This result is rather disappointing, considering the objective of the generative model is to sample more high-reward modes. In this context, the authors need to discuss the effect of the proposed backward policies on the performance of the forward policy in more detail.
2. In Figure 4, lower rewards achieved by FragGFN and FragGFN SA seem questionable because FragGFN covers a much larger chemical space, as seen from Figure 2, and so is more likely to find molecules with higher rewards. Has the poor performance of FragGFN originated from insufficient training? Or is it because of the use of SeH proxy as a reward, which may have a severe generalization problem for unseen molecules? This should be justified in the revision.
3. In Section 4.1, what is the rewarding strategy used for multi-objective optimization (FragGFN SA, SynFlowNet SA, SynFlowNet QED)? The possible reward structures could be Multi-objective GFlowNet (MOGFN) or multiplication-based rewards.
4. Regarding the metric Reward/Novelty trade-off, it is possible that improving the novelty for SynFlowNet may be simply due to the use of the Enamine building blocks, while REINVENT has been trained with ChEMBL. Thus, the high similarity of the molecules generated by REINVENT to ChEMBL is not unexpected and would be reduced if training it with the Enamine REAL. The authors need to address this issue.

---

> ### Author Response · Authors · 2024-11-20
> **Author response (1/2)**
>
> We thank the reviewer for valuable feedback. We will respond to each of the reviewer's questions individually.
>
> >  Increasing the number of BBs slows down training linearly that may cause a computational burden when dealing with a large amount of BBs.
>
> While the linear slow down of training with increasing number of BBs is worrying, an action space of >200k BBs (which is the current size of the Enamine BBs) can be easily handled by the current implementation of SynFlowNet. With multi-processing, we were able to train a model with 200k BBs and 10k steps, with a batch size of 64 in ~6 hours. Additionally, by looking at the history of Enamine's building blocks set, it has never increased by a lot more than 25k BBs at any given year [(Enamine Real Compounds -- Youtube)](https://www.youtube.com/watch?v=Vn5Z2nFhXL4&t=1495s). Projecting this 10 years forward, the size of the set would reach ~600k in 10 years, which can be managed by today's SynFlowNet without any change. If the set of BBs was to grow faster at any point, we could consider hierarchical action spaces to handle that more efficiently.
>
> > 1. the authors need to discuss the effect of the proposed backward policies on the performance of the forward policy in more detail.
>
> The last column of Table 2 shows that backward policies trained using the Maximum Likelihood criterion or using the REINFORCE algorithm lead to the discovery of a higher number of high-reward modes compared to a free backward policy only trained to match the trajectory balance criterion, and perform similarly to a uniform backward policy, two strategies often employed in the literature [1]. Figure A.6 in Appendix further supports the claims and shows how the number of discovered high-reward modes progresses with training under the different backward policies, which looks favorable for a REINFORCE policy with a positive entropy coefficient.  Crucially, as explained in Section 4.4, training the backward policy with the proposed objectives does not primarily attempt to improve the performance of the forward policy, but aims at insuring the forward-backward flow consistency in the MDP. In a synthesis DAG we cannot guarantee that a backward-constructed trajectory ends in $s_0$ (i.e. that leads back to a building block -- the retrosynthesis problem). This shortcoming would cause the model to sample from a different, biased distribution than from the Boltzmann distribution defined by the reward function $R$. In Table 2, we thus show that training the backward policy according to any of our proposed strategies dramatically increases the ability of the model to return to $s_0$ when traversing the MDP in the backward direction, thus ensuring flow-consistency and correcting that bias. We have made this point clearer in the updated manuscript to emphasize that the real benefit of our proposed training scheme for the backward policy is preserving the correct flow in the MDP *while* retaining the ability to discover diverse and high-reward modes. We believe that this opens up interesting opportunities for future work by showing an example where a data-driven, ML-based solution is employed as a solution to an MDP design challenge.
>
> > 2. Has the poor performance of FragGFN originated from insufficient training? Or is it because of the use of SeH proxy as a reward, which may have a severe generalization problem for unseen molecules?
>
> Indeed, we were also expecting better performance for the fragment models. We hypothesise that the observed performance can be due to the following reason: to offer a fair comparison between the models in terms of chemical space covered, FragGFN models were trained using the fragments obtained from decomposing the Enamine building blocks used by SynFlowNet. The space of obtained fragments is large (~250 fragments) and leads to a very large state space (as estimated in Figure 2), therefore finding high reward modes in such a large space can be difficult. We use a similar training budget to SynFlowNet and the experiment shows the advantage of SynFlowNet in finding high-reward modes faster. We have now clarified this in the manuscript.
>
> > 3. what is the rewarding strategy used for multi-objective optimization (FragGFN SA, SynFlowNet SA, SynFlowNet QED)?
>
> The strategy that we used was the multiplication of rewards, which has now been clarified in the manuscript.

---

> > ### Comment · Reviewer_6JHQ · 2024-11-25
> >
> > Thank the authors for their efforts to update the paper. They have addressed all my comments. However, I still have a concern about scaling the building block size. While the Enamine library increases, as the author estimated, many more building blocks and reaction templates can be readily assembled like the Enamine. The following paper claims that a trillion-scale chemical library can be made with millions of building blocks (ACS Med. Chem. Lett. 2023, 14, 4, 466–472). Thanks to the advanced automated synthesis methods, this size may not increase linearly.

---

> > > ### Author Response · Authors · 2024-11-25
> > > **Thank you for your positive feedback**
> > >
> > > Thank you for the feedback and further question!
> > >
> > > Indeed, the size of enumerated chemical spaces grows exponentially. This for example would represent a significant scaling challenge for likelihood-based molecule generators, which would soon have to be trained against exponentially growing datasets. SynFlowNet, being built on a sequential decision making formulation task, already allows to tackle this exponential growth efficiently [1, 2]. So the space as a whole may be growing exponentially, but the GFlowNet only has to adapt to the growth in the set of building blocks, which as we have pointed to, on Enamine's side tends to grow linearly and can be managed by today's SynFlowNet.
> > >
> > > It is true that one possibility would be to use products themselves as additional building blocks. Such an approach has been tackled by [3], where two products can be assembled simultaneously and later merged in the generation process. This strategy is also compatible with SynFlowNet, and we leave it to future work to explore the implementation options.
> > >
> > > Finally, one consideration to keep in mind for the integration of additional building blocks or for the use of products as building blocks is that molecules resulting from automated synthesis methods such as eXplore are much larger than the Enamine building blocks, which have an average number of heavy atoms equal to 14 (see Figure 6 in [4]) and therefore would not be considered in the current action space of SynFlowNet. That is because we want synthesis pathways to start from small building blocks, so that the size of the final product is drug-like. We show in Figure 2 of our manuscript that with such building blocks, we still explore very large chemical spaces, up to $10^{15}$.
> > >
> > > Let us know if there are further questions!
> > >
> > > ---
> > >
> > > References
> > >
> > > [1]: Bengio, E., Jain, M., Korablyov, M., Precup, D., & Bengio, Y. (2021). Flow network based generative models for non-iterative diverse candidate generation. Advances in Neural Information Processing Systems, 34, 27381-27394.
> > >
> > > [2]: Bengio, Y., Lahlou, S., Deleu, T., Hu, E. J., Tiwari, M., & Bengio, E. (2023). Gflownet foundations. The Journal of Machine Learning Research, 24(1), 10006-10060.
> > >
> > > [3]: Gao, W., Mercado, R., & Coley, C. W. (2021). Amortized tree generation for bottom-up synthesis planning and synthesizable molecular design. arXiv preprint arXiv:2110.06389.
> > >
> > > [4]: Neumann, A., Marrison, L., & Klein, R. (2023): Relevance of the Trillion-Sized Chemical Space “eXplore” as a Source for Drug Discovery. ACS Medicinal Chemistry Letters, 14(4), 466-472

---

> > > > ### Comment · Reviewer_6JHQ · 2024-11-26
> > > >
> > > > Thank you for the detailed reply. This reviewer will raise the score!

---

> > > > > ### Author Response · Authors · 2024-11-26
> > > > >
> > > > > Thank you for updating the score!

---

> ### Author Response · Authors · 2024-11-20
> **Author response (2/2)**
>
> > 4. it is possible that improving the novelty for SynFlowNet may be simply due to the use of the Enamine building blocks, while REINVENT has been trained with ChEMBL?
>
> To fully answer this question and the suggestion of reviewer AADH, we have retrained SynFlowNet with building blocks derived from ChEMBL molecules and performed the same analysis as in Figure 6 ([edit]: now Figure 5). We decomposed 70k randomly sampled ChEMBL molecules into building blocks using a retrosynthesis software and used these to train SynFlowNet. We observe that although SynFlowNet was trained with a ChEMBL-dervied chemical space, it manages to preserve the novelty reported with Enamine building blocks, with an increase in average similarity from 0.43 to 0.49. This experiment thus strengthens the argument that SynFlowNet discovers more novel regions of the chemical space, as opposed to models relying on likelihood-based priors such as REINVENT, which show difficulty in departing from their training distribution. We report these results in Section B.1, Table A.2 and Figure A.5, and in the table below.
>
>
> | Method                       | sEH proxy (↑)       | Diversity (↑)       | SA (↓)            | AiZynth (↑) | QED (↑)          | Mol. weight (↓) | ChEMBL Similarity (↓) |
> |------------------------------|---------------------|---------------------|-------------------|-------------|------------------|-----------------|-----------------------|
> | REINVENT                    | 0.91 ± 0.01        | 0.68 ± 0.02        | 2.19 ± 0.02       | 0.95        | 0.57 ± 0.04     | 429.78 ± 23.02 | 0.64 ± 0.02          |
> | SynFlowNet (L = 3)      | 0.92 ± 0.01   | 0.79 ± 0.01    | 2.92 ± 0.10   | 0.65    | 0.59 ± 0.02 | 365.23 ± 2.42 | 0.43 ± 0.01      |
> | **SynFlowNet ChEMBL (L = 3)**   | **0.91 ± 0.02**        | **0.80 ± 0.01**        | **2.68 ± 0.18**       | **0.67**        | **0.68 ± 0.03**     | **342.67 ± 8.00**  | **0.49 ± 0.01**          |
>
>
> ---
> References
>
> [1] Malkin, Nikolay et al., “Trajectory Balance: Improved credit assignment in GFlowNets.” CoRR, 2201.13259, 2022

---

> > ### Comment · Reviewer_6JHQ · 2024-11-25
> >
> > Thanks for clarifying my concern with the additional experiment. Indeed, the result supports SynFlowNet's superior diversity and novelty performance over REINVENT.

---

### Official Review · Reviewer_f18i · 2024-11-03

**Soundness:** 4
**Presentation:** 4
**Contribution:** 3
**Rating:** 8
**Confidence:** 4

**Summary:**

It was a pleasure to review this paper on synthesizability-constrained molecular generation using GFlowNets. Overall, I felt the paper was strong and focuses on an important application area of generative models, it was well-written, and generally builds well on previous work in the field and introduces some novelty. In my review below, I focus on aspects and unresolved questions that, if addressed, could enhance the paper. I summarize my main concerns as: (1) in its current state, the work is not reproducible, and (2) the evaluation metrics and comparisons to prior work used in evaluating SynFlowNet are not always the most relevant for molecular design, and do not tell us much about how this method would actually perform in a realistic setting. There is room for improvement here. I think this paper has the potential to be a really strong paper if it addresses the concerns below.

**Strengths:**

* The background and related work section was strong, and covered the previous work in this domain well.
* The paper, including the appendix, was well-structured and easy to follow.
* There are more strengths, but I mention them below amongst some of the specific questions I had, to put the questions in context.

**Weaknesses:**

* I searched throughout the paper and in the submission, and could not find any code to accompany the paper. Given that the authors lamented the lack of publicly-available code for two previous works in their paper, I would have expected a greater emphasis on publishing the code here, as it would also help with the reproducibility of this work. To me this is unacceptable for a computer science conference submission in 2024, so please let me know if I simply missed it, and I can raise my score (I ranked it below the acceptance threshold because of this, and would update my score if the code was made available as it was overall a good paper).
* Throughout the paper, the authors repeat that one of the key advantages of SynFlowNet is its ability to generate more diverse candidates than other methods (namely, REINVENT and SyntheMol). However, I argue that diversity is only a meaningful metric in molecular generative models if the types of molecules being generated are from the accessible molecular space; if the generated molecules are diverse but not accessible, then the model is not very practically useful. That being said, it was good the the authors evaluated synthesizability using AiZynthFinder, but an "easy" experiment that the authors could have also done that would add a lot of value is to consider molecular rediscovery benchmarks. That is, can the model also find known, previously-synthesized binders; celecoxib rediscovery is a widely-used standard benchmark for this, but there are also more challenging targets. Such rediscovery metrics are a nice alternative to experimental validation which is not necessarily possible on the review timeline. To summarize, I would like to know that SynFlowNet can generate diverse and *known* synthesizable molecules, not just diverse *predicted* synthesizable molecules, because at the end of the day CASP tools are not a replacement for experiment.
* I found the lack of comparison to other methods for synthesizability-constrained molecular generation a weakness. The authors mentioned that the models of Gottipati and Horwood & Noutahi were not publicly available, and thus focused on comparison to SyntheMol and REINVENT. While these comparisons were interesting and certainly well-motivated, I would have also liked to know how SynFlowNet compares to other synthesizability-constrained molecular generation models which *do* have publicly available codebases, like MoleculeChef or SynNet. As it stands, we do not know which model is best, because SyntheMol has not done any comparisons to those models either.
* In the Appendix, the authors state that 300K oracle calls were used both to train REINVENT and SynFlowNet, however, this is an extreme number of oracle calls; can the authors justify this? REINVENT demonstrates good performance after only 10K oracle calls for most objectives, so are 300K oracle calls simply what is needed for good performance in SynFlowNet? I would like to see the analogous experiments and comparisons/results with only 10K oracle calls, which is the standard number of oracle calls used for comparing molecular generative models.

**Questions:**

* I could not find how $s_0$ is initialized, nor how important this is (i.e., how different initializations here impact the performance of SynFlowNet). Can the authors clarify? Does it correspond to a molecular fingerprint/embedding of the target molecule, is it random noise, or something else?
* One of the contributions of this paper is stated as "we propose one of the first attempts at training the backward policy in a GFlowNet with a separate objective from for the forward policy, which we show to correct backward-generated trajectories and improve sample quality and diversity" but I am not sure I got that. Which experiment was it exactly that demonstrated the new backward policy led to improved sample quality/diversity? Are these results in the paper? If so this needs to be clarified.
* How sample efficient is SynFlowNet? I would have liked to see an analysis of this, which is really important for molecular generation tasks as we are almost always operating in the low data regime and data/sample efficiency is key.
* How does SynFlowNet perform when optimizing for targets that are highly symmetric, e.g., molecules with a Cn rotation axis, where n$\ge$2? Has this been evaluated? This is something that is a known weakness of other synthesizability-constrained molecular generation frameworks (e.g., SynNet).
* What are the failure modes of SynFlowNet? This should be more thoughtfully analyzed, as all models have failure modes and it is important to understand them to help future research build on the results of SynFlowNet.
* How challenging would it be to expand SynFlowNet beyond bi-molecular reactions? I am not suggesting this should have been done here, but asking more from curiousity and wondering if this is a future direction the authors are aiming to explore in future work.

Minor:
* Are the points in Figure 6 (last panel) shuffled for plotting? It was not clear to me if there are red points under the blue ones or vice-versa. This should be clarified as the results from this plot are informing the conclusions.
* The references are poorly formatted, this should be fixed.
* gsk3$\beta$ should be GSK3$\beta$ (weird capitalization in the Appendix).

---

> ### Author Response · Authors · 2024-11-19
> **Author response (1/n)**
>
> Dear Reviewer f18i,
>
> Thank you for reviewing our manuscript and for the insightful comments. We have now updated our manuscript based on your suggestions. Below we address each question individually.
>
> > could not find any code to accompany the paper
>
> We have now added a link in our manuscript to the anonymised Github repository for this project: [https://anonymous.4open.science/r/synflownet-01A3](https://anonymous.4open.science/r/synflownet-01A3/README.md).
>
> > I would like to know that SynFlowNet can generate diverse and known synthesizable molecules, not just diverse predicted synthesizable molecules
>
> We have added results showing SynFlowNet's performance in rediscovery tasks (celecoxib and aripiprazole), which we also show in the table below. Being synthesis-constrained, the expressivity of the model is reduced compared to models constructing molecules at an atom or fragment level (such as those in [1]). The performance of SynFlowNet in such rediscovery tasks is confined by the particular set of building blocks and reactions made available to the model. If a particular target molecule cannot be synthesized from these components, it will remain inaccessible for SynFlowNet. We verified whether the two molecules tested for this task have analogues in the REAL space 2024-02 using SpaceLight [2] and found that only aripiprazole is present, explaining the low rediscovery score of celecoxib (closest analogue found has similarity 0.68. Note that the REAL space is assembled from different reactions to the ones used in SynFlowNet, therefore the 0.68 analogue might not be reachable).
>
> | Task                   | Reward (↑)         | SA (↓)           |
> |------------------------|--------------------|------------------|
> | Aripiprazole rediscovery | 0.90 ± 0.00       | 2.19 ± 0.00      |
> | Celecoxib rediscovery  | 0.48 ± 0.01       | 2.47 ± 0.04      |
>
> >  I would have also liked to know how SynFlowNet compares to other synthesizability-constrained molecular generation models which do have publicly available codebases
>
> Our goal in the paper was to focus on comparisons to four families of models:
>
> - **Different algorithm using the same MDP:** This analysis is aimed at RL models such as those of Gottipati et al. and Horwood & Noutahi et al.. However, the code to reproduce their method using our own MDP was not available. To offer comparison to the RL counterpart of SynFlowNet, we thus implemented Soft Q-learning, a well known, strong RL baseline similar to the algorithms employed in Gottipati et al. and Horwood & Noutahi et al.
> - **Same algorithm (GFlowNet) but using a different MDP**: For this comparison, we focused on Fragment GFlowNet, which uses the same training procedure but samples molecules according to a different action space. To maximise fairness, we also re-generated the fragment library used by this model to closely match SynFlowNet's chemical space.
> - **A likelihood-based model and a strong baseline from the literature**: REINVENT
> - **Search-based synthesisability-constrained model**: SyntheMol
>
> We believe that these represent a strong and diverse set of empirical comparisons which allow to analyse the strengths and limitations of the presented approach from diverse angles.
>
> We also have edited the manuscript to incorporate the suggested comparison to SynNet. In Table A.4. we show new results for SynFlowNet trained on all optimization tasks performed in SynNet, and we report top-k performance. We compare these results with that of SynNet extracted from the original paper [3] and find that the two models are comparable: SynFlowNet achieves better performance for GSK3$\beta$ (0.862 vs. 0.815) and SynNet scores higher with JNK3 (0.719 vs. 0.710).
>
> > are 300K oracle calls simply what is needed for good performance in SynFlowNet?
>
> SynFlowNet can learn to generate molecules of high reward with less oracle calls (see the newly-added Table A.6 ([edit]: now Table A.5) in Appendix), however we continue to train the model with 300k oracle calls to let it discover more modes of the distribution captured by the reward function. As shown in the bottom of Figure 5 ([edit]: now Figure 4) in the paper, the most important gains in performance happen in the first 20% of the training run, but SynFlowNet then continues to discover new high reward modes for many additional steps with diminishing returns. How long the model should be trained thus depend on how expensive the oracle function is to query (see next question) and the level of accuracy that is targetted w.r.t the sampling distribution of the model.

---

> ### Author Response · Authors · 2024-11-19
> **Author response (2/n)**
>
> > How sample efficient is SynFlowNet?
>
> We now introduced a study of SynFlowNet’s performance on the PMO benchmark, which shows improvement over the Fragment GFlowNet (Table A.6 [edit]: now Table A.5). Notably, SynFlowNet does not rely on a pretrained prior, which is the case of the high-performing models in the PMO benchmark, and learns to generate molecules purely from the distribution of the reward function. Although requiring more oracle calls, SynFlowNet offers a more unbiased solution on the sampling problem, while (as shown in the last panel of Figure 7 ([edit]: now Figure 6)), it is hard for REINVENT, which uses a prior, to leave the distribution of that prior. Generative models based exclusively on sampling from a reward function, such as SynFlowNet, are mostly designed for relatively inexpensive oracle functions, such as medium-accuracy docking simulations, or high-throughput screening platforms which can produce hundreds of thousands of readouts each weeks or months.
>
> | Task  | Method     | AUC   |
> |-------|------------|---------------|
> | GSK3β | REINVENT   | 0.865 ± 0.043 |
> |       | SynNet     | 0.789 ± 0.032 |
> |       | FragGFN    | 0.651 ± 0.026 |
> |       | SynFlowNet | 0.691 ± 0.034 |
> | DRD2  | REINVENT   | 0.943 ± 0.005 |
> |       | SynNet     | 0.969 ± 0.004 |
> |       | FragGFN    | 0.590 ± 0.070 |
> |       | SynFlowNet | 0.885 ± 0.027 |
>
> >I could not find how $s_0$ is initialized, nor how important this is
>
> On lines 205/206 we state “In more detail, each trajectory starts from an empty molecular graph which is followed by a building block sampled from AddFirstReactant”. This explains that $s_0$ is an empty molecular graph.
>
> >Which experiment was it exactly that demonstrated the new backward policy led to improved sample quality/diversity?
>
> The last column of Table 1 ([edit]: now Table 2) shows that backward policies trained using the Maximum Likelihood criterion or using the REINFORCE algorithm lead to the discovery of a higher number of high-reward modes compared to a free backward policy only trained to match the trajectory balance criterion, and perform similarly to a uniform backward policy, two strategies often employed in the literature [4]. Crucially, as explained in Section 4.4, training the backward policy with the proposed objectives does not primarily attempt to improve the performance of the forward policy, but aims at insuring the forward-backward flow consistency in the MDP. In a synthesis DAG we cannot guarantee that a backward-constructed trajectory ends in $s_0$ (leads to a building block, i.e. the retrosynthesis problem), which will cause the model to sample from a slightly different, biased distribution than from the Boltzmann distribution defined by the reward function $R$. In Table 1 ([edit]: now Table 2), we also show that training the backward policy according to any of our proposed strategies dramatically increases the ability of the model to return to $s_0$ when traversing the MDP in the backward direction, thus ensuring flow-consistency and correcting that bias. We have made this point clearer in the updated manuscript to emphasize that the real benefit of our proposed training scheme for the backward policy is preserving the correct flow in the MDP *while* retaining the ability to discover diverse and high-reward modes. We believe that this opens up interesting opportunities for future work by showing an example where a data-driven, ML-based solution is employed as a solution to an MDP design challenge.
>
> > How does SynFlowNet perform when optimizing for targets that are highly symmetric, e.g., molecules with a Cn rotation axis, where n$\geq$2?
>
> We have tested this against a symmetrical target "CC(N)CCc1cccc(C(=O)CCC(=O)CCC(=O)c2cccc(CCC(C)N)c2)c1" and found that SynFlowNet was able to discover it (top-1 reward=1.0).
>
> >What are the failure modes of SynFlowNet?
>
> We have now added Section B.2 "Outlook" explaining the failure modes of SynFlowNet and potential ideas for improvement left for future work.
>
> > How challenging would it be to expand SynFlowNet beyond bi-molecular reactions? I am not suggesting this should have been done here, but asking more from curiousity and wondering if this is a future direction the authors are aiming to explore in future work.
>
> Expanding SynFlowNet beyond bi-molecular reactions is an interesting direction which we have actually considered. Tri-molecular reactions can be accounted for by additional masking based on the location of substructure match. For now, we have successfully extended SynFlowNet to work with scaffold decoration, which vouched for the flexibility of the codebase and way in which the action space is structured.
>
> > Are the points in Figure 6 (last panel) shuffled for plotting?
>
> No, the points are not shuffled. The points have increased transparencies from top to bottom.
>
> > The references are poorly formatted, this should be fixed. gsk3$\beta$ should be GSK3$\beta$
>
> Thanks for pointing this out, we have revised these.

---

> > ### Author Response · Authors · 2024-11-19
> > **References**
> >
> > [1] Brown, Nathan et al., “GuacaMol: Benchmarking Models for de Novo Molecular Design.” J. Chem. Inf. Model 2019;59(3):1096-1108
> >
> > [2] SpaceLight, BioSolveIt https://www.biosolveit.de/spacelight-a-spotlight-on-the-analog-hunter-for-chemical-spaces/
> >
> > [3] Gao, Wenhao et al., "Amortized tree generation for bottom-up synthesis planning and synthesizable molecular design", 2022: https://arxiv.org/abs/2110.06389
> >
> > [4] Malkin, Nikolay et al., “Trajectory Balance: Improved credit assignment in GFlowNets.” CoRR, 2201.13259, 2022

---

> ### Comment · Reviewer_f18i · 2024-11-21
>
> Thank you to the authors for the thoughtful revisions.
>
> I see that a codebase has been added and many of my suggested revisions addressed (at least in the comments and through additions to the appendix), and increased my score from a 5 to a 6. I will have a deeper look this Saturday to see if the added experiments are also discussed in the main text, as they are quite interesting.
>
> Thank you for also adding the sample efficiency results in addition to the rediscovery results. I think that it is okay that SynFlowNet does not outperform REINVENT in sample efficiency, as years of engineering have gone into making REINVENT efficient and it is also not especially designed for synthesizability-constrained generation; but it is always good to have things to aim for in future work.
>
> Finally, I would possibly suggest to replace Figure 3 currently in the main text to make room for Section B.1.1 in the main text, as that is more interesting; in other words, I recommend moving the example synthesis pathway to the appendix as IMO it does not add as much as a discussion on molecular rediscovery (though, Figure 3 is a very nice figure).

---

> > ### Author Response · Authors · 2024-11-22
> > **Thank you for your positive review**
> >
> > We thank the reviewer for their appreciation of our work and once they take a deeper dive, we welcome further thoughts that would make the paper a strong ICLR contribution.
> >
> > We agree that the take-away from the rediscovery results is stronger than the former Figure 3 with the synthesis pathway example, therefore we included them in the main text. We have also updated a new version where the added experiments from rebuttals are now better discussed in the main text.

---

> > > ### Comment · Reviewer_f18i · 2024-11-25
> > >
> > > Thanks to the authors for all the hard work on improving the paper and addressing not only my suggestions but also those of the other reviewers.
> > >
> > > I can see that the paper is now very strong, and a very good contribution to the field.
> > >
> > > I have updated my score accordingly.

---

> > > > ### Author Response · Authors · 2024-11-25
> > > >
> > > > Thank you for updating the score and for your feedback!

---

### Author Response · Authors · 2024-11-19
**General response**

We thank all reviewers for their time and effort in providing feedback and constructive criticism on our work. In our individual responses to each reviewer we address all comments and questions. In our global response below we highlight the most important changes we've made to our manuscript. All changes are highlighted in blue in the revised version of the paper.

1. We uploaded the code accompanying the paper: [https://anonymous.4open.science/r/synflownet-01A3](https://anonymous.4open.science/r/synflownet-01A3/README.md)
2. Following the suggestions of reviewers f18i and cHNB, we included sample efficiency results on the PMO benchmark and added evaluations of SynFlowNet on rediscovery tasks.
3. Following the suggestions of reviewers 6JHQ and AADH, we support our novelty claims with additional experiments of training SynFlowNet with ChEMBL-derived building blocks, which show that we preserve novelty scores.
4. We clarified statements with regards to the scalability of our approach, the benefits of the proposed backward policies, the limitations of the model and we better explain various experimental details and design choices.

We thank all reviewers again for their feedback and we remain available for further questions!

---

### Meta-Review · Area_Chair_hcsL · 2024-12-22

**Metareview:**

In this work, authors introduce SynFlowNet, a generative model that combines GFlowNets with reaction-based Markov Decision Processes to generate synthetically accessible drug-like molecules. The key technical innovation is incorporating forward synthesis constraints directly into the generative process by using commercially available reactants and documented chemical reactions. The model introduces novel backward policy training strategies to ensure generated molecules have valid synthetic routes.

The paper demonstrates several important scientific findings. First, SynFlowNet achieves better diversity in generated molecules compared to baseline methods like REINVENT while maintaining high rewards, as shown through extensive experiments with both proxy models and docking scores (Reviewer f18i). Second, the model scales effectively to large building block sets of over 200k compounds, though with expected computational overhead (Reviewer 6JHQ). Third, the backward policy training innovations help ensure generated molecules have valid synthetic routes back to building blocks, addressing a key challenge in synthesis-constrained generation (Reviewers AADH and 6JHQ).
The paper's key strengths include: 1) A well-engineered solution to the critical problem of ensuring synthetic accessibility in molecular generation, validated through multiple metrics and external retrosynthesis tools, 2) Comprehensive empirical evaluation including ablation studies, comparison to strong baselines, and demonstration on practical drug discovery tasks, 3) Clear practical relevance demonstrated through integration with experimental fragment screening data.

The initial submission had some limitations around reproducibility and evaluation completeness, but these were thoroughly addressed during review. The authors added code, conducted additional experiments on molecular rediscovery tasks and sample efficiency (addressing Reviewer f18i's concerns), and validated the novelty claims by testing with ChEMBL-derived building blocks (addressing Reviewers 6JHQ and AADH's concerns). The revised evaluation provides strong evidence for the method's capabilities.

The authors were highly responsive during review, strengthening the evaluation to conclusively demonstrate the benefits of their approach. While there remain some scaling challenges with large building block sets, the current capabilities are sufficient for many practical applications. The combination of technical innovation, thorough evaluation, and clear practical relevance makes this a strong contribution to the field. I recommend acceptance because the paper makes a significant technical contribution to synthesizable molecular generation while maintaining high standards of empirical validation, an important problem that has received lesser attention in the molecule generation community.

**Additional Comments On Reviewer Discussion:**

The discussion revealed broad reviewer consensus on the paper's merits after revisions, with all reviewers increasing their scores. The authors' careful attention to reviewer feedback and willingness to conduct substantial additional experiments provides further confidence in the quality of the work.

---

### Decision · Program_Chairs · 2025-01-22

Accept (Spotlight)